# On the Accurate Estimation of Information-Theoretic Quantities from Multi-Dimensional Sample Data

**DOI:** 10.3390/e26050387

**Published:** 2024-04-30

**Authors:** Manuel Álvarez Chaves, Hoshin V. Gupta, Uwe Ehret, Anneli Guthke

**Affiliations:** 1Stuttgart Center for Simulation Science, Cluster of Excellence EXC 2075, University of Stuttgart, 70569 Stuttgart, Germany; 2Hydrology and Atmospheric Sciences, The University of Arizona, Tucson, AZ 85721, USA; 3Institute of Water and River Basin Management, Karlsruhe Institute of Technology (KIT), 76131 Karlsruhe, Germany

**Keywords:** information theory, non-parametric estimation, entropy, mutual information, Kullback–Leibler divergence, relative entropy, data, binning, kernel density estimation, *k*-nearest neighbors, *k*-NN

## Abstract

Using information-theoretic quantities in practical applications with continuous data is often hindered by the fact that probability density functions need to be estimated in higher dimensions, which can become unreliable or even computationally unfeasible. To make these useful quantities more accessible, alternative approaches such as binned frequencies using histograms and *k*-nearest neighbors (*k*-NN) have been proposed. However, a systematic comparison of the applicability of these methods has been lacking. We wish to fill this gap by comparing kernel-density-based estimation (KDE) with these two alternatives in carefully designed synthetic test cases. Specifically, we wish to estimate the information-theoretic quantities: entropy, Kullback–Leibler divergence, and mutual information, from sample data. As a reference, the results are compared to closed-form solutions or numerical integrals. We generate samples from distributions of various shapes in dimensions ranging from one to ten. We evaluate the estimators’ performance as a function of sample size, distribution characteristics, and chosen hyperparameters. We further compare the required computation time and specific implementation challenges. Notably, *k*-NN estimation tends to outperform other methods, considering algorithmic implementation, computational efficiency, and estimation accuracy, especially with sufficient data. This study provides valuable insights into the strengths and limitations of the different estimation methods for information-theoretic quantities. It also highlights the significance of considering the characteristics of the data, as well as the targeted information-theoretic quantity when selecting an appropriate estimation technique. These findings will assist scientists and practitioners in choosing the most suitable method, considering their specific application and available data. We have collected the compared estimation methods in a ready-to-use open-source Python 3 toolbox and, thereby, hope to promote the use of information-theoretic quantities by researchers and practitioners to evaluate the information in data and models in various disciplines.

## 1. Introduction

### 1.1. The Promise of Information Theory

Entropy, mutual information, and Kullback–Leibler (KL) divergence are fundamental concepts of information theory [1]. Originally introduced in the field of communication [2], information theory has now found uses in a diverse set of disciplines, including artificial intelligence [3], Earth and environmental science [4], experimental design [5], neuroscience [6], and finance and economics [7]. Its wide-ranging applications stem from its solid foundation in probability theory. By analyzing the probability distributions associated with the variables in a given problem, information theory can determine the nature and extent of their relationships. These relationships may exhibit linearity or non-linearity, depending on the specific system involved [8]. In essence, information theory can unveil the intricate connections hidden within complex systems.

### 1.2. Definition of Information-Theoretic Quantities

For the initial definitions and notation, we follow Cover and Thomas [1] and MacKay [3]. In the discrete case, consider *x* the outcome of a random variable *X*, which takes on one of the set of possible values in the alphabet $A_{X}=\left\{a_{1},a_{2},\ldots ,a_{n}\right\}$ having probabilities $P_{X}=\left\{p_{1},p_{2},\ldots ,p_{n}\right\}$ with the probability mass function (PMF) $P\left(X=a_{i}\right)=p_{i}$, $p_{i}\geq 0$, and $\sum_{a_{i}\in A_{X}}P\left(X=a_{i}\right)=1$. An example of an alphabet is the 27 characters in a random English document (letters from a to z and the space character) [3].

The information content of the outcome *x* is defined as:(1)$$h\left(x\right)=-\log_{2}P\left(x\right),$$
with the unit of “bit” due to the base two of the logarithm. The *entropy* H(X) of the random variable *X* is defined to be the average information content of every possible outcome. Therefore:(2)$$H\left(X\right)=-\sum_{x\in A_{X}}P\left(x\right)\log_{2}P\left(x\right).$$

The relative entropy or *KL divergence* $D_{KL}$ between two probability distributions P(x) and Q(x) that are defined over the same alphabet $A_{X}$ is:(3)$$D_{KL}\left(P\;||\;Q\right)=\sum_{x}P\left(x\right)\log_{2}\frac{P(x)}{Q(x)}.$$

A simple interpretation of the KL divergence is a measure of the inefficiency of assuming that the distribution is *Q* when the true distribution is *P*.

Introducing a second outcome *y* of a random variable *Y* from a different alphabet $A_{Y}$, the joint entropy of the two random variables *X* and *Y* is:(4)$$H\left(X,Y\right)=-\sum_{xy\in A_{X}A_{Y}}P\left(x,y\right)\log_{2}P\left(x,y\right).$$

Entropy is additive only for independent random variables:(5)H(X,Y)=H(X)+H(Y)⇔P(x,y)=P(x)P(y).

From this result, *mutual information* I(X;Y) can be introduced as the KL divergence between the joint PMF P(x,y) and the product of the marginal PMFs P(x) and P(y):(6)$$\begin{array}{cc} I(X;Y) & =D_{KL}\left(P\left(x,y\right)\;||\;P\left(x\right)P\left(y\right)\right)\\ \; & =\sum_{x\in A_{X}}\sum_{y\in A_{Y}}P\left(x,y\right)\log_{2}\frac{P(x,y)}{P(x)P(y)} \end{array}$$

Mutual information is positive and symmetric, i.e., I(X;Y)=I(Y;X) and I(X;Y)=0, only when Equation (Equation 5) is true. Using the rules of conditional probabilities, namely P(x,y)=P(y)P(x|y), one arrives at the more common definition for mutual information:(7)$$\begin{array}{cc} I(X;Y) & =H(X)-H(X|Y)\\ \; & =H(X)+H(Y)-H(X,Y) \end{array}$$

As per Equation (Equation 7), mutual information measures the average reduction in uncertainty about *X* that results from the knowledge of *Y*.

In his seminal paper, Shannon [2] introduced the concept of differential entropy specifically for continuous random variables. In this context, consider a random variable *X* with values *x* in $R^{d}$ space and a probability density function (PDF) p(x) whose support is a set X. Its differential entropy is given by Equation (Equation 8):(8)$$H\left(X\right)=-\int_{X}p\left(x\right)\log p\left(x\right)\;dx$$

In this equation, we have made some changes in the notation to emphasize our interest in the continuous setting. The alphabet $A_{X}$ becomes the support set X. PMFs denoted as *P* become PDFs denoted as *p*. The $\log_{2}$ is substituted simply by the log, which is understood to be the natural logarithm, also changing the units in which information is measured from “bit” to “nat” or the natural unit of information and further differentiating between the discrete and continuous case. One bit is equivalent to log(2) nats.

Similarly, the equations for mutual information and KL divergence can also be adapted for continuous applications (Equations (Equation 9) and (Equation 10), respectively):(9)$$D_{KL}\left(p||\;q\right)=\int_{X}p\left(x\right)\log\left(\frac{p(x)}{q(x)}\right)dx$$
(10)$$I\left(X;Y\right)=\int_{Y}\int_{X}p\left(x,y\right)\log\left(\frac{p(x,y)}{p(x)\;p(y)}\right)dx\;dy$$

Note that Equation (Equation 9) requires the support of q(x) to be equal to or larger than the support of p(x) because the integral is over the support of the latter.

As will be described later in Section 1.3, the definition of differential entropy for continuous variables simply replaces a sum with an integral, but this is not a well-defined operation. For the cases of KL divergence and mutual information, the issue of discretization is partially avoided because the argument in the logarithm is a ratio of probabilities over the same space [3], but this is not the case for differential entropy.

### 1.3. The Challenge of Estimating Information-Theoretic Quantities

Beirlant et al. [9] provide a comprehensive review of various common approaches to estimating differential entropy. In this section, we will follow their classification while noting that these methods can also be extended to the cases of KL divergence and mutual information.

Generally, estimation methods using sample data can be divided into three categories. First are plug-in estimates, which can be further divided into resubstitution and integral estimates. Resubstitution estimates calculate a density for each data point in a sample, and then, the integration term in Equations (Equation 8)–(Equation 10) is exchanged for a sum over densities. Integral estimates use a density representation of the total sample to perform numerical integration of Equations (Equation 8)–(Equation 10). Plug-in estimates are often associated with *kernel density estimation* and will be explained further in Section 2.1.

Second are estimates based on sample-spacings, or histograms as they are more commonly known and from this time forward referred to as *binning*. They share the density estimation step used in plug-in estimates, but differ in some other aspects, which will be explored further in Section 2.2.

It becomes obvious that these two categories of estimation methods involve an initial step of density estimation from samples [10] before computing the desired information-theoretic quantities. However, density estimation itself poses challenges [11,12], and common techniques of density estimation often perform poorly on high-dimensional data (d>3). Kernel density methods are often used to address high-dimensional problems, but Joe [13] studied these methods for entropy estimation, concluding that they perform well only when the number of dimensions of the multivariate data is small.

The third category, namely estimates based on *nearest neighbor* distances, differ from the previous two methods in that they do not require an explicit density estimate. Density can be estimated using distance-based methods [14], but this is not necessary in this case. Hence, they promise to be relatively straightforward to implement and robust, which makes them a good candidate to compare with more traditional density-based estimates. Nearest neighbor distance-based estimates will be detailed further in Section 2.3.

While the literature for each of the listed methods has been well established [9,15] and more recent approaches to density estimation have proven to perform well in higher dimensions [16], in practice, we often find binning to be the most commonly adopted technique [17,18,19,20] without further justification. We hypothesize that this preference for a single method with known deficiencies, at least in higher dimensions, is due to the lack of a systematic comparison of methods, and due to a perceived initial hurdle to implementation.

### 1.4. Contribution and Outline of This Study

In this study, we compare kernel density estimation, binning, and the nearest neighbor approaches for estimating information-theoretic quantities in a practical setting. In a comprehensive and systematic effort, we compare these three most widely recognized non-parametric estimation methods in terms of (1) their theoretical derivation, (2) their sample size requirements, and (3) their accuracy. While our focus primarily centers on estimating entropy, KL divergence, and mutual information due to their foundational significance, it is important to acknowledge that additional measures such as conditional entropy can be derived from these fundamental quantities. To this end, we have designed a set of eight synthetic scenarios to test each estimation method on data coming from different uni- and multi-variate distributions, ranging up to ten dimensions. Usually, when a method is introduced, there is a comparison of its performance against other methods; however, typically, this comparison is limited to two particular algorithms [21], and often, the comparison is performed between algorithms of the same family, e.g., *k*-NN [22]. Also, the comparison of different methods is typically challenging due to the absence of a standardized set of practices across cases [23], making it difficult to ensure comparability between them.

Finally, we provide a free toolbox that collects these methods and test cases, to promote the use of these concepts in a wide range of scientific and practical applications.

The remainder of this article is structured as follows: In Section 2, we introduce the non-parametric estimation methods studied in this article, discuss their properties and hyperparameters, and describe how they are applied for the computation of the information-theoretic quantities of interest. Section 3 first presents the general design of the test cases in which the estimators are evaluated. Each test case is then introduced with a description of the specific probability distribution we sample from, and closed-form solutions to the quantities of interest, if available. The results are presented and analyzed with the help of “evaluation matrix plots”, which show the absolute and relative performance of each estimator across the different quantities. Finally, in Section 4, we summarize our findings and discuss how they may guide future applications of information theory in domain-specific workflows.

## 2. Definition and Theoretical Description of Estimators

This section describes the three non-parametric estimation methods investigated in this study and discusses specific aspects for each of them.

### 2.1. Kernel Density Estimation (KDE)

Introduced as the Parzen–Rosenblatt window method [24,25], the KDE method consists of estimating a PDF based on kernels as weights, with the kernel being a non-negative window function. The density p(x) at a point *x* is estimated as:(11)$$\hat{p}\left(x\right)=\frac{1}{n}\sum_{i=1}^{n}K\left(u\right),$$
where
(12)$$u=\frac{{\left(x-x_{i}\right)}^{⊺}\Sigma^{-1}\left(x-x_{i}\right)}{h^{2}}$$
and *n* is the total number of samples, *K* is a multivariate kernel function, $x_{i}={\left[x_{1,i},x_{2,i},\ldots ,x_{d,i}\right]}^{⊺}$ is a *d*-dimensional vector of samples, Σ is the covariance matrix of the samples, and *h* is a smoothing parameter. For the results presented further in this paper, we chose to always use a multivariate Gaussian kernel function and the Silverman bandwidth estimate, as suggested by Moon et al. [26].

The multivariate Gaussian kernel takes the form:(13)$$K\left(u\right)=\frac{1}{{\left(2\pi \right)}^{d/2}h^{d}\det{\left(\Sigma \right)}^{1/2}}e^{-u/2},$$
where *u* was previously defined in Equation (Equation 12) and *h* is Silverman’s bandwidth estimate [11], given by:(14)$$h={\left(\frac{n\left(d+2\right)}{4}\right)}^{-1/\left(d+4\right)}.$$

Using the density estimate from Equation (Equation 11), entropy, mutual information, and KL divergence can be calculated directly as:(15)$$H\left(X\right)=-\frac{1}{n}\sum_{i=1}^{n}\log\left(\hat{p}\left(x_{i}\right)\right)$$
(16)$$D_{KL}(\hat{p}\left|\right|\hat{q})=\frac{1}{n}\sum_{i=1}^{n}\log\left(\frac{\hat{p}\left(x_{i}\right)}{\hat{q}\left(x_{i}\right)}\right)$$
(17)$$I\left(X;Y\right)=\frac{1}{n}\sum_{i=1}^{n}\log\left(\frac{\hat{p}\left(x_{i},y_{i}\right)}{\hat{p}\left(x_{i}\right)\hat{p}\left(y_{i}\right)}\right)$$

In this form, these can be called resubstitution estimates of each quantity. Additionally, if numerical integration using the kernel PDF is used to approximate the result of Equations (Equation 8)–(Equation 10), this is known as an integral estimate [9]. For example, Equation (Equation 8) would be written as:(18)$$H\left(X\right)=-\int_{X}\hat{p}\left(x\right)\log\hat{p}\left(x\right)\;dx$$

Both resubstitution and integral estimates are plug-in estimates.

### 2.2. Binning

The binning method relies on obtaining an estimate of the density using a histogram [27,28]. Given an origin at $x_{0}$ and a bin width of delta (Δ), the bins of the histogram are set to the intervals $\left[x_{0}+m\Delta ,x_{0}+\left(m+1\right)\Delta \right]$ for a set number of positive and negative integers of *m*. Then, considering the total number of observations *n* and the number of observations $c_{i}$ in the same bin as $x_{i}$, the frequency estimate at $x_{i}$ in the histogram can be written as:(19)$$\hat{f}\left(x_{i}\right)=\frac{c_{i}}{n\Delta }.$$

In this case, we have chosen to use *f* instead of *p* to distinguish between a frequency count and a probability density estimate. To obtain probabilities from this frequency estimate, one must account for the specified bin width p(x)=Δf(x). In Equation (Equation 19), the parameter Δ controls the amount of smoothing applied to the frequency estimate, similar to the bandwidth used in Section 2.1. To ease the readability, we have chosen to show equations that use a bin spacing that is uniform, but every bin could have a particular size, independent of the rest $\left(\Delta_{i}\right)$.

In terms of probability, we follow the notation of Cover and Thomas [1] in which $X^{\Delta }$ represents the quantized (binned) version of a random variable. Then, following their derivation, the entropy of this quantized version can be written as:(20)$$H\left(X^{\Delta }\right)=-\sum \Delta \hat{f}\left(x_{i}\right)\log\hat{f}\left(x_{i}\right)-\sum \hat{f}\left(x_{i}\right)\Delta \log\Delta ,$$
where ∑Δf(x)=∫p(x)=1. The second term on the right-hand side of Equation (Equation 20) is a correction factor due to the bin spacing of the chosen quantization of *x*.

As KL divergence and mutual information evaluate ratios between distributions, this correction factor cancels out, and then, their expressions can be written similarly to Equations (Equation 16) and (Equation 17):(21)$$D_{KL}(\hat{f}\left|\right|\hat{g})=\frac{1}{n}\sum_{i=1}^{n}\log\left(\frac{\hat{f}\left(x_{i}\right)}{\hat{g}\left(x_{i}\right)}\right)$$
(22)$$I\left(X;Y\right)=\frac{1}{n}\sum_{i=1}^{n}\log\left(\frac{\hat{f}\left(x_{i},y_{i}\right)}{\hat{f}\left(x_{i}\right)\hat{f}\left(y_{i}\right)}\right)$$

Here, the second PDF $\hat{g}\left(x_{i}\right)$ is estimated using the same binning scheme as $\hat{f}\left(x_{i}\right)$.

Even though the correction factor cancels out for the previous two equations, the final result does depend on the particular binning scheme chosen. A particular choice of binning controls the trade-off between a resulting histogram that has too much detail (“undersmoothing”) or a histogram that has too little detail (“oversmoothing”) with respect to the true distribution. The resulting estimate of the KL divergence or mutual information, therefore, also depends on the selection of a particular binning scheme. Because of this reason, several rules-of-thumb have been developed over time for the selection of an “optimal” bin width or binning scheme.

#### 2.2.1. Rules-of-Thumb for Bin Width Selection

For the purpose of obtaining a good representation of the underlying distribution of the data, there are several methods to estimate an adequate number of bins or, more specifically, the bin width Δ to be used when building a histogram. Some rules-of-thumb exist to estimate Δ for one-dimensional data. In the context of multi-dimensional data, methods for optimal bin width estimation have been proposed, but they often require the use of combinatorial methods [29] or the solution of an optimization problem [30]. Therefore, we chose the simpler approach of estimating Δ independently for each dimension of the data and built a multi-dimensional histogram based on the estimated Δs, meaning that binning is uniform in each dimension, but not across dimensions.

*Sturges’ rule:* The width of each bin is the base two logarithm of the number of samples in the data (*n*), $\Delta =\log_{2}\left(n\right)+1$. With this estimate of the number of bins, there is an inherent assumption that the data follows a normal distribution [31].

*Scott’s rule:* The width of each bin is proportional to the standard deviation (σ) of the data, and inversely proportional to the cube root of the number of samples (*n*), $\Delta =3.49\;\sigma n^{-1/3}$. Although there is still an assumption that the data follow a normal distribution, this assumption is not as strong as with Sturges’ rule [27].

*Freedman and Diaconis’ rule:* The width of each bin is proportional to the interquartile range (IQR) of the data, and inversely proportional to the number of samples (*n*), $\Delta =2\;\mathrm{IQR}/n^{1/3}$. Although this is similar to Scott’s rule, in using the IQR, this estimator for the bin width is more robust to outliers of non-normal distributions [32].

#### 2.2.2. The Quantile Spacing Approach

Taking a different perspective on optimal binning for estimating entropy from a sample, Gupta et al. [10] recently introduced the Quantile Spacing (QS) approach for the case where *X* is a one-dimensional continuous random variable and the mathematical form for the distribution of the data-generating process is unknown. The approach is based in the assumption that the PDF can be approximated as piecewise constant on the intervals between quantile locations where the values for the quantiles have been determined from the sample *S*.

The method assumes that the PDF can be approximated as piecewise constant on the intervals between quantiles $Z=\{z_{0},z_{1},z_{2},\ldots ,z_{N_{Z}}\}$, where $N_{Z}$ represents the total number of quantiles and $z_{0}=x_{min}$ and $z_{N_{Z}}=x_{max}$. Using the quantiles *Z*, the PDF can be approximated as:(23)$$p\left(x\right)\approx \hat{p}\left(x|Z\right)=p_{j-1}^{j}=\frac{K}{\Delta_{j}}\;\mathrm{for}\;z_{j-1}\leq X\leq z_{j}\;\mathrm{and}\;j=1,\ldots ,N_{Z},$$
where $\Delta_{j}=z_{j}-z_{j-1}$ and $K=\frac{1}{N_{Z}}$. From this definition of the quantiles, the entropy estimate is given by the sum of the individual entropies of each uniform distribution across quantiles. More specifically, the estimate depends on the logs of the spacings between quantiles and is defined by the average of these values:(24)$${\hat{H}}_{\hat{p}}\left(X|Z\right)=\frac{1}{N_{Z}}\cdot \sum_{i=1}^{N_{Z}}\log\left(N_{Z}\cdot \Delta_{j}\right)$$

To determine the empirical quantiles, $N_{k}$ sample subsets are sampled without replacement from the full sample *S*, each of size $N_{Z}-1$. These subsamples are then sorted to obtain an estimate of the quantile locations, thereby obtaining $N_{k}$ estimates of each quantile $z_{j}$. Finally, the average of the value of the location is taken to obtain $\hat{Z}=\left\{x_{\min},{\hat{z}}_{1},{\hat{z}}_{2},\ldots ,{\hat{z}}_{N_{Z}-1},x_{\max}\right\}$.

As described, the QS approach has two hyperparameters: $N_{Z}$ and $N_{k}$. The choice of $N_{Z}$ is suggested to be 25% of the total number of points in the sample $N_{S}$, as it minimizes the bias in the estimate for the normal, exponential, and log-normal distributions [10]. $N_{k}$ is recommended to be a large number as a greater number of subsamples and repetitions more accurately estimate the correct value for each quantile, with $N_{k}=500$ found to be optimal [10].

### 2.3. k-Nearest Neighbors (k-NN)

The *k*-nearest neighbors (*k*-NN) algorithm is a non-parametric supervised learning method originally developed for classification and regression [33]. It aims at local function approximation by assuming similarity between neighboring sample points. Since the distance of a sample point to its *k*-nearest neighbors can be interpreted as a local density estimate, it is not a surprise that also *k*-NN-based estimates of information-theoretic quantities have been proposed. However, the charm of these estimates is precisely that they do *not* require an explicit evaluation of a probability density estimate.

The *k*-NN-based estimator for entropy was introduced by Kozachenko and Leonenko [34] and serves as the basis for *k*-NN-based estimations of KL divergence and mutual information:(25)$$\hat{H}\left(X\right)=\psi \left(N\right)-\psi \left(k\right)+\log\left(c_{1}\left(d\right)\right)+\frac{d}{N}\sum_{i=1}^{N}\log\left(\rho_{k}^{d}\left(i\right)\right),$$
where ψ is the digamma function defined as the logarithmic derivative of the gamma function ($\frac{d}{dz}\log\left(\Gamma \left(z\right)\right)$). *N* is the total number of points in the sample. *k* is a hyperparameter specifying the number of nearest neighbors used in the estimate. $c_{1}\left(d\right)$ is the volume of a *d*-dimensional unit ball with *d* being the number of dimensions of the sample. $\rho_{k}^{d}\left(i\right)$ is the distance between $x_{i}$ and its k^*th*^ nearest neighbor. In calculating the distance, the length between two points (x and y) is given by a *p*-norm function, where p≥1, as follows:(26)$${∥x-y∥}_{p}=\left(\right|x_{1}-y_{1}{|}^{p}+|x_{2}-y_{2}{|}^{p}+\ldots +|x_{n}-y_{n}{{|}^{p})}^{\frac{1}{p}},$$
with a suggested p=2, i.e., using the Euclidean norm for the entropy estimate [34].

Kozachenko and Leonenko [34] demonstrated that the suggested estimator has square consistency for any number of dimensions, meaning that, as the size of the sample increases, the mean-squared error of the estimate tends to 0:(27)$$\lim_{n\to \infty }E\left\{{\left(\hat{H}-H\left(f\right)\right)}^{2}\right\}=0,$$
where $\hat{H}$ is the estimate of H(f). Additionally, Delattre and Fournier [35] studied the bias and variance of the *k*-NN estimator for entropy, suggesting that it is unbiased up to even a strong form of consistency, but variance increases as the number of dimensions *d* becomes higher. Such a type of consistency has also been claimed for the KDE estimator described in Section 2.1 by Ahmad and Lin [36].

The estimator for KL divergence was proposed by Wang et al. [23]. Considering *p* and *q* as continuous PDFs in $R^{d}$, let $\left\{X_{1},\;\ldots \;,X_{n}\right\}$ and $\left\{Y_{1},\;\ldots \;,Y_{m}\right\}$ be independent and identically distributed (i.i.d.) *d*-dimensional samples drawn from *p* and *q*. Then, the proposed estimator for KL divergence is:(28)$${\hat{D}}_{KL\;n,m}\left(p||q\right)=\frac{d}{n}\sum_{i=1}^{n}\log\left(\frac{\nu_{k}\left(i\right)}{\rho_{k}\left(i\right)}\right)+\log\left(\frac{m}{n-1}\right)$$
where $\rho_{k}\left(i\right)$ is the distance between $X_{i}$ and its *k*-NN in ${\left\{X_{j}\right\}}_{j\neq i}$ and $\nu_{k}\left(i\right)$ is the distance between $X_{i}$ and its *k*-NN in $\left\{Y_{j}\right\}$. The authors also demonstrate the mean-squared consistency of the estimator, as in Equation (Equation 27). p=2 or the Euclidean norm is also suggested to calculate the distance.

Finally, the estimator for mutual information was proposed by Kraskov et al. [21]:(29)$$\hat{I}\left(X;Y\right)=\psi \left(k\right)-\frac{1}{N}\sum_{i=1}^{N}E\left[\psi \left(n_{i,x}+1\right)+\psi \left(n_{i,y}+1\right)\right]+\psi \left(N\right)$$
where $n_{i,x}$ and $n_{i,y}$ are the number of neighbors in the X and Y spaces inside a radius given by the distance up to the k^*th*^ nearest neighbor in the joint X-Y space. The authors require that the distance be calculated using p=∞, i.e., the infinite or maximum norm.

Gao et al. [37] analyzed the properties of the proposed *k*-NN estimator of mutual information, finding it consistent, as it was with the previous two estimators. Further, they determined upper bounds of the rate of convergence in the estimate as a function of the dimensions of the two random variables involved.

To illustrate the somewhat abstract idea of using *k*-NN for estimating information-theoretic quantities, assume we wish to determine the mutual information between two univariate random variables *X* and *Y*. The algorithm for Equation (Equation 29) then works as follows: for each pair $\left(x_{i},y_{i}\right)$ in the data, find its k^*th*^ nearest neighbor in the joint *X*-*Y* space, then count the number of neighbors inside a radius of distance $\rho_{i}^{k}/2$ in the *X* and *Y* spaces. Because the maximum norm is used (p=∞), the neighbors are found strictly inside a row (X) and a column (Y) of width $\rho_{i}^{k}$. The sketch in Figure 1 serves to illustrate this procedure. In the case of multiple dimensions, the rows and columns for *X* and *Y* become hyper-stripes [15].

As the *k*-NN equations do not require an explicit evaluation of a probability density estimate, they look nothing like the canonical formulae shown in Equations (Equation 8)–(Equation 10). Nevertheless, an estimate of density at a point $x_{i}$ can be calculated using *k*-NN:(30)$${\hat{p}}_{k}\left(x_{i}\right)=\frac{k}{N-1}\cdot \frac{1}{c_{1}\left(d\right)\cdot \rho_{k}^{d}\left(i\right)}$$

This *k*-NN-based density estimation could be used as a plug-in estimate for entropy, for example, but this is not recommended, as this density estimate has some issues, which will be explored further in Section 3.4.2.

## 3. Comparison of Estimator Performance in Test Cases of Increasing Complexity

### 3.1. Test Case Design

We investigated the efficiency and accuracy of the different estimators described in Section 2 on a range of test cases summarized in Table 1. For each test case, we generated a sample of a fixed size from a given distribution and test the estimators’ ability to quantify the three information-theoretic measures entropy, KL divergence, and mutual information.

We varied the sample size in order to reveal how data-hungry the different methods are in producing a reasonably good estimate of the true measure for the underlying distribution. Note that the difficulty is not only to have a large enough sample for the estimation method to work reliably, but, by its nature, a limited sample can only represent its corresponding theoretical distribution to a limited extent. Hence, we expect the estimates based on small sample sizes to be of mixed quality partly because they are not representative enough of the true distribution. We repeatedly drew samples of a certain size and report the average and the confidence interval of the estimates. We expect the performance and the confidence of the estimators to generally increase with the sample size due to this representation effect; yet, it remains to be observed how the different estimators behave for the different sample sizes and how they compare with each other in terms of accuracy and efficiency. The robustness of an estimator might be increased by bootstrapping; however, this will not resolve the issue of (non-)representativeness. It is beyond the scope of this study to design and recommend the best-suited bootstrapping approaches for the individual estimators; further, initial tests not shown here have confirmed the expectation that the variability resulting from bootstrapping is dominated by the much larger sampling variability, which is accounted for in this study, and hence, we neglected the effect of bootstrapping in this investigation.

The complexity of the test cases increases with respect to dimensionality (ranging from one to ten) and skewness (ranging from symmetrical to highly skewed) of the chosen distributions.

For the simplest 1D cases, there is no dependent variable to calculate mutual information on; so, in Cases 1 to 3, we only investigated the estimators’ performance in calculating entropy and KL divergence. We deliberately started our investigation with low dimensions and well-behaved distribution shapes such that we can rely on analytical reference solutions for the information-theoretic quantities. Then, for cases of increased complexity, analytical equations no longer exist (or only for some quantities of interest), and we have to switch to computationally heavy numerical integration to serve as a reference for the true value of the target quantity (Cases 3, 5, and 6).

### 3.2. Choice of Hyperparameters

Each estimation method needs the tuning of certain hyperparameters or making methodological choices. In the case of binning, the hyperparameter is the bin size Δ, and we investigated the differences in performance due to the four different methods to select the bin size presented in Section 2.2. In histogram-based methods, it is important to clarify how bins with zero probability, or empty bins, are handled because of the log function in all quantities of interest. In our specific application, where Equations (Equation 20)–(Equation 22) are sums, bins that have zero probability are excluded before the summation operation takes place. The hyperparameters of the QS approach are fixed to the recommended values of the original proposal [10] as described in Section 2.2.2.

For KDE, we show the difference between performing a resubstitution estimate as opposed to an estimate through numerical integration. In both cases, a kernel representation of the PDF of the data is created using a Gaussian kernel and Silverman’s bandwidth estimate, as described in Section 2.1. Then, the resubstitution estimate is obtained by averaging over the kernel PDF evaluated at all available data points in the sample, while the integration estimate is obtained by taking the expected value. Specifically, for entropy, Equations (Equation 15) and (Equation 18) show how these two processes are different.

As for *k*-NN, we investigated the effect of varying the number of neighbors used, *k*, between 1 and 15. The value of *k* was cut off there because, for larger values (e.g., k=50), the results showed very similar behavior to k=15, but with larger variability in small sample sizes.

### 3.3. Implementation

The sample sizes ranged between 100 and 100,000 samples, with the upper limit being reduced to 5000 in some cases related to the computationally expensive KDE-based estimator. The test case procedure was repeated across 300 different fixed seeds for random sampling to show confidence intervals related to the effect of sampling variability on an estimate.

Numerical integration as a reference when there is no analytical solution was achieved through the QUADPACK [38] (quad and nquad for higher dimensions) interface in *SciPy* [39]. For the integration scheme, default settings were used where both the absolute and relative error tolerances were set to $1.49\times 10^{-8}$. Integration limits were set according to the true support of the underlying distribution.

Each of the estimators was implemented as a Python 3 [40] function. The binned estimators use the histogram, histogram-dd, and digitize functions from *NumPy* [41], the latter two used for multi-dimensional histograms with uneven and even binning widths, respectively. The KDE and *k*-NN estimators rely on the Gaussian KDE and the *k*-d Tree data structures available in *SciPy* [39], respectively.

The code for the setup of the test cases presented here to allow for reproducing our results and, e.g., future testing of further alternative methods is available at the *Data Repository of the University of Stuttgart (DaRUS)*. Further, all of the estimation methods have been collected as part of the open-source *UNITE toolbox*. The published toolbox itself contains the functions for the estimators to be applied to any case study of interest.

### 3.4. Case 1: 1D, Uniform Distribution

To begin our evaluation of the different estimation methods presented in this paper, we start with the uniform distribution. In the continuous case of the uniform distribution, the total mass of the distribution lies uniformly between two arbitrary bounds. As such, it is the maximum entropy distribution for a random variable *X* under no other constraints [3]. If the bounds of the uniform distribution are *a* and *b*, where b>a, then the PDF is:(31)$$p\left(x\right)=\left\{\begin{array}{cc} \frac{1}{b-a} & \mathrm{for}\;a\leq x\leq b\\ 0 & \mathrm{for}\;x<a\;\mathrm{or}\;x>b \end{array}\right.$$

#### 3.4.1. Analytical Reference Solution

The entropy of a uniform distribution p(x) is given by
(32)H(X)=log(b−a).

If q(x) is a second uniform distribution with bounds *c* and *d* where $\left(a,b\right)\subset \left(c,d\right)$, the KL divergence between the two distributions p(x) and q(x) is:(33)$$D_{KL}\left(p\;||\;q\right)=\log\left(\frac{d-c}{b-a}\right).$$

Figure 2 shows the PDFs of the distributions used to estimate entropy and KL divergence, together with the estimators’ analytical reference value. For KL divergence, the approximating distribution (q(x) in red) must have greater or equal support for the true distribution (p(x) in black), which is the case for the example shown.

#### 3.4.2. Density Estimation

For illustrative purposes, we start off by showing the ability of the different estimation methods to make an estimate of this one-dimensional uniform distribution. Figure 3 shows the true underlying PDF and the three PDFs derived by each method applied for density estimation, i.e., Equations (Equation 11), (Equation 19), and (Equation 30). The solid line for each method represents the mean of the estimated density across 300 random seeds of 10,000 samples of the uniform distribution, and the density was calculated for 500 evenly spaced points between 0.0 and 2.5 along the *x* axis. Additionally, the $2.5^{th}$ and $97.5^{th}$ percentile of the density estimated at each point is shown as the shaded area in the same color as the mean for all methods.

As expected, KDE gives a very accurate density estimation within the limits of the underlying distribution, but the effect of the Gaussian kernel is apparent close and outside of the limits. A boxcar-function-based kernel might be most suitable for this particular example, but said recommendation becomes more difficult when the data are higher dimensional; therefore, the Gaussian kernel is the default option [11]. Although binning produces estimates with higher variance than KDE, it seems to be the most suitable method for this example. The *k*-NN-based method was not explicitly designed to perform density estimation, and the example in Figure 3 graphically shows the large variance that the method has even in a 1D case [35]. The variance is affected by the number of neighbors used for density estimation; for smaller values of *k*, the variance is much higher than for larger values of *k*. In Figure 3, k=50 was used. This is a much larger value than what is typically recommended, as we will see later, but appears adequate for density estimation, and larger values of *k* would indeed reduce the variance further. Nevertheless, the uniform distribution remains challenging due to the steps at the limits of the distribution, where only density estimation through binning is able to identify the “hard” cutoff limits.

Recall that density estimation is not the primary goal of our comparison; instead, we are interested in the estimation quality of the three information-theoretic quantities. Only the KDE and binning methods rely on this initial density estimation step; *k*-NN skips this step and is only shown here for a better intuition about the characteristics of this approach.

#### 3.4.3. Estimation Results

The estimation results for the case of the uniform distribution are shown in Figure 4. The estimates of the three methods (columns) for the two quantities entropy and KL divergence (rows) are shown as absolute values in nats on the left-hand axes, whereas the relative errors with respect to the reference solution can be read off from the right-hand axes. Note that the axis scaling is consistent within each row (i.e., between estimates for a specific quantity), but not necessarily across the rows (i.e., between the different quantities).

This first test case serves as an intuitive starting point for the interpretation of Figure 4, as the results for all further test cases follow the same template.

The KDE estimator gives a greatly biased result when estimating entropy due to the effect of using a Gaussian kernel to approximate a uniform PDF (as seen in Figure 3). This is because the Gaussian kernel widens the true support of the uniform distribution, and therefore, it leads to an overestimation of entropy. Integration reduces this effect and also reduces the variance of the estimate in smaller sample sizes.

Binning underestimates entropy for sample sizes up to 1000 and then yields very accurate and precise estimates. This observation is well in line with the theory, since a too-small-to-be-representative sample will not be perfectly uniform and, hence, the true entropy value of the maximum-entropy distribution is approached from below with increasing sample size. The QS approach typically underestimates entropy when compared to a histogram-based approach and one of the considered “rules-of-thumb”.

The *k*-NN estimator for entropy gives a similarly accurate result as binning, but with a much higher variance. As discussed in the context of density estimation, the *k*-NN method does not rely on any smoothing via kernels or bins, and hence, the sampling variability hits its result hard. As expected, this variability is reduced with increasing number of neighbors; however, a smaller number of neighbors yields more accurate results for smaller sample sizes.

Estimating KL divergence directly using the KDE estimator gives a result very close to the truth even in small sample sizes, with an increase in the number of samples only reducing the variance. Across different samples, the ratio between the true and the approximating distribution seems to stay consistent, leading to this result. Numerical integration leads to a worse outcome as the effect of the Gaussian kernel becomes more apparent (both distributions are first approximated, and then, their ratio is assessed, “doubling” the smearing effect of the kernel).

For the binning estimator, we see a “reversal effect” in the estimate for KL divergence where Sturges’ rule initially computes a larger KL divergence, but starts to produce results that are lower than Scott’s and FD’s rule at 500 samples. The reason is the resulting number of bins obtained from the different methods described in Section 2.2.1: across larger sample sizes, Scott’s and FD’s rules result in a very similar number of bins with FD’s estimate being usually slightly larger, and both being larger than Sturges’ rule. This situation is reversed in smaller sample sizes, with Sturges’ rule producing the largest number of bins.

The *k*-NN approach yields the best results for KL divergence with a single neighbor (k=1). While, again, the variance is high for small sample sizes, the estimate is highly accurate across all sample sizes. Obviously, a one-to-one comparison of estimated densities is preferable for an accurate estimate of KL divergence, as also seen for KDE (a direct estimate being much superior to integration). While intermediate numbers of neighbors perform poorly, the highest number investigated here (k=15) comes close to the accuracy of k=1, but fails to reach it. It is also remarkable that *k*-NN-based estimation of $D_{KL}$ shows very little variation across sample sizes, again emphasizing that a local estimate of the ratio between the two distributions is less affected by unrepresentative sampling or distribution reconstruction.

### 3.5. Case 2: 1D, Normal Distribution

The normal or Gaussian distribution is the second of our test cases and also serves as the basis for the applications in higher dimensions, where the estimators will be tested on multivariate normal distributions (Section 3.10 and Section 3.11). In the most basic one-dimensional case, the PDF of the normal distribution is:(34)$$p\left(x\right)=\frac{1}{\sigma \sqrt{2\pi }}e^{-\frac{1}{2}{\left(\frac{x-\mu }{\sigma }\right)}^{2}}$$

The normal distribution is often denoted as $N(\mu ,\;\sigma^{2})$, where μ is the expected value of the distribution and σ is the standard deviation.

#### 3.5.1. Analytical Reference Solution

Given the PDF in Equation (Equation 34), the entropy of the normal distribution is:(35)$$H\left(X\right)=\frac{1}{2}\ln\left(2\pi \sigma^{2}\right)+\frac{1}{2}$$

Given two normal distributions, where $p\left(x\right)=N\left(\mu_{p},\;\sigma_{p}^{2}\right)$ and $q\left(x\right)=N\left(\mu_{q},\;\sigma_{q}^{2}\right)$, the KL divergence between these two distributions is:(36)$$D_{KL}\left(p||\;q\right)=\frac{1}{2}\left\{{\left(\frac{\sigma_{p}}{\sigma_{q}}\right)}^{2}+\frac{{\left(\mu_{q}-\mu_{p}\right)}^{2}}{\sigma_{q}^{2}}-1+\ln\left(\frac{\sigma_{q}^{2}}{\sigma_{p}^{2}}\right)\right\}$$

Figure 5 shows the distributions used in this application case.

#### 3.5.2. Estimation Results

The estimation results for the case of the normal distribution are shown in Figure 6. Estimating the density of a normal distribution with a Gaussian kernel is a prime task for KDE, so the relative errors in estimating the two information-theoretic quantities are expected to be small. Across all sample sizes and quantities, a direct evaluation of the KDE estimator performs better than performing numerical integration over the kernel representation. The latter gives additional importance to the overall shape of the distribution, which is an approximation in both cases, rather than focusing on the specific points, which support the kernel representation. This can be seen more clearly for KL divergence, where ratios are considered. Only at large sample sizes, where the overall representation of the underlying distribution is very close to the true distribution, numerical integration gives a result that is almost as good as direct estimation.

All binning methods, with the exception of Sturges’ rule, give very similar results, which are highly accurate for large sample sizes. This is expected as the rules-of-thumb were all derived with more or less strict assumptions on the distribution being Gaussian. The bin width given by Scott’s works best because it is optimal for random samples of normally distributed data, in the sense that it minimizes the integrated mean-squared error of the density estimate [27]. Neither FD’s and Sturges’ estimate follow this same property and allow for a more flexible representation of the underlying distribution. On normally distributed data and in large sample sizes, FD’s rule equally matches Scott’s rule, but Sturges’ rule overestimates entropy and underestimates Kullback–Leibler divergence because it calculates a larger Δ and a smaller number of bins. The QS approach does not assume normally distributed data (and hence, also cannot benefit from that knowledge); as in the previous case, it tends to slightly underestimate entropy, but its performance is generally very similar to the other binning methods. Remember that it is only derived for estimating entropy, not KL divergence or other multivariate quantities.

The results of using *k*-NN to estimate entropy and KL divergence appear consistent with those of the previous case: for small sample sizes, the bias decreases with decreasing *k*, but the variance increases. However, for sample sizes >1000, the bias and variance almost diminish; also, for any choice of hyperparameter and sample size, the relative error of the entropy estimates is generally small, with the largest relative error being ≈6%. This demonstrates the effectiveness of this estimation method when applied to normally distributed data. For KL divergence, again, k=1 is the best choice of hyperparameter, and given a specific *k*, the mean estimates of KL divergence are very consistent across sample sizes.

### 3.6. Case 3: 1D, Normal Mixture Distribution

To exploit the statistical properties of normal distributions while making them more flexible in fitting non-normal real-word data sets, it is common to use weighted mixtures of normal distributions in approximation tasks. Such a mixture offers the possibility to test the estimators presented in this paper in a setting where the data come from a distribution that is not uni-modal and more skewed. Further, the main interest in evaluating a mixture of normal distributions is that a normal mixture model is a universal approximator of densities, in the sense that any smooth density can be approximated with any specific nonzero amount of error by a normal mixture model with enough components [33]. The PDF of a normal mixture distribution is:(37)$$p\left(x\right)=\sum_{i=1}^{n}\omega_{i}\;N\left(\mu_{i},\;\sigma_{i}^{2}\right),$$
where $\omega_{i}$ is the individual weight of each of the *n* components of the mixture, with their individual expected values $\mu_{i}$ and standard deviations $\sigma_{i}$, and $\sum_{i=1}^{n}\omega_{i}=1$.

#### 3.6.1. Numerical Reference Solution

Because of the logarithm of a sum of exponential functions, the entropy of a mixture of normal distributions cannot be calculated [42]. The same applies to KL divergence. Therefore, numerical integration was used to obtain the reference value presented in the results.

Figure 7a shows the distribution used for this application case. This is an equally weighted mixture of the two normal distributions $N(-2.5,\;2.5^{2})$ and $N(2.5,\;1^{2})$.

#### 3.6.2. Estimation Results

The estimation results of the mixture of normal distributions are shown in Figure 8 and they are very similar to those of the single normal distribution in the previous case. KDE again performs well due to the usage of a Gaussian kernel. Given the two peaks of the distribution as seen in Figure 7a, for a low number of samples, the kernel density at specific points is underestimated, and therefore, the entropy is overestimated. This is the opposite as in Case 2 (pure normal distribution), in which small samples tend to over-represent a narrower high-probability range of the underlying distribution, which leads to an underestimation of entropy. In both cases, direct estimation gives a very accurate result for sample sizes larger than 1000; integration struggles much more in the case of the normal mixture distribution.

For the binning estimator, the results are also very similar to the previous case: both Scott’s and FD’s estimate perform very well; the QS approach converges to their result from below; Sturges’ rule struggles even more than before due to an inadequate number of bins for both estimating entropy and KL divergence. Scott’s rule appears to be best to calculate an adequate number of bins, particularly for KL divergence where the estimate based on said rule appears to consistently match the true value across all sample sizes.

*k*-NN-based estimation behaves similarly to all previous one-dimensional cases. Here, once again, values of k=3 or k=5 give an estimate of entropy that is very close to the true value while having a smaller variance than k=1. k=15 gives the estimate with the smallest variance, but largest bias for small sample sizes. For estimating KL divergence, again, k=1 performs best and k=15 approaches this result (but still shows a small bias even for large sample sizes).

### 3.7. Case 4: 2D, Normal Distribution

For extending Case 2 to a normal distribution with *d* dimensions, we use the notation $N_{d}\left(\mu ,\;\Sigma \right)$, where $\mu \in R^{d}$ is a *d*-dimensional mean vector and $\Sigma \in R^{d\times d}$ is a d×d covariance matrix, where Σ is positive-definite. For this distribution, the PDF is:(38)$$p\left(x\right)=\frac{1}{{\left(2\pi \right)}^{d/2}{\left(\det\left(\Sigma \right)\right)}^{1/2}}\exp\left(-\frac{1}{2}{\left(x-\mu \right)}^{⊺}\Sigma^{-1}\left(x-\mu \right)\right)$$

#### 3.7.1. Analytical Reference Solution

The entropy of this multivariate normal distribution is given by [1]:(39)$$H\left(X\right)=\frac{1}{2}\ln\left({\left(2\pi e\right)}^{d}\cdot \det\left(\Sigma \right)\right)$$

The KL divergence between two *d*-dimensional normal distributions $p\left(x\right)=N_{d}\left(\mu_{p},\;\Sigma_{p}\right)$ and $q\left(x\right)=N_{d}\left(\mu_{q},\;\Sigma_{q}\right)$ [43]:(40)$$\begin{array}{c} D_{KL}\left(p||\;q\right)=\frac{1}{2}\left[\ln\left(\frac{\det\left(\Sigma_{q}\right)}{\det\left(\Sigma_{p}\right)}\right)+\mathrm{Tr}\left(\Sigma_{q}^{-1}\Sigma_{p}\right)+{\left(\mu_{q}-\mu_{p}\right)}^{⊺}\Sigma_{q}^{-1}\left(\mu_{q}-\mu_{p}\right)-d\right] \end{array}$$

In particular, for d=2, the parameters for the normal distribution become a 2×1 vector for the means and a 2×2 matrix for the covariance. These parameters can be written as:$$\begin{array}{cc} \mu ={\left[\begin{array}{cc} \mu_{0} & \mu_{1} \end{array}\right]}^{⊺} & \Sigma =\left[\begin{array}{cc} \sigma_{0}^{2} & \rho \;\sigma_{0}\;\sigma_{1}\\ \rho \;\sigma_{0}\;\sigma_{1} & \sigma_{1}^{2} \end{array}\right] \end{array}$$

This notation is similar to that of Equation (Equation 34) with the addition of ρ being the Pearson correlation coefficient between $X_{0}$ and $X_{1}$. Using this parametrization, the expression for mutual information for the bi-variate normal distribution becomes:(41)$$I\left(X_{0},\;X_{1}\right)=-\frac{1}{2}\log\left(1-\rho^{2}\right)$$

This means that, for a bivariate normal distribution, there is an exact relationship between the correlation coefficient ρ and mutual information [44].

Figure 9a shows a bivariate normal distribution $p\left(x_{1},\;x_{2}\right)=N_{2}\left(\mu_{p},\;\Sigma_{p}\right)$, and Figure 9b shows the same, with a second approximating distribution $q\left(x_{1},\;x_{2}\right)=N_{2}\left(\mu_{q},\;\Sigma_{q}\right)$ in red. The parameters for both distributions are:$$\begin{array}{cc} \mu_{p}={\left[\begin{array}{cc} -2 & 0 \end{array}\right]}^{⊺} & \mu_{q}={\left[\begin{array}{cc} 0 & 0 \end{array}\right]}^{⊺}\\ \; & \\ \Sigma_{p}=\left[\begin{array}{cc} 1 & -0.5\\ -0.5 & 1 \end{array}\right] & \Sigma_{q}=\left[\begin{array}{cc} 5 & 0\\ 0 & 1 \end{array}\right] \end{array}$$

Additionally, in $\Sigma_{p}$, the value of ρ is −0.5 and can be directly used in Equation (Equation 41). This result is also shown in Figure 9a.

#### 3.7.2. Estimation Results

This initial case in two dimensions starts to introduce the challenges of density estimation in higher dimensions. The estimation results are shown in Figure 10. More specifically, for KDE-based estimation, direct evaluation of a kernel representation of the probability distribution is very computationally expensive for sample sizes larger than 25,000, and the waiting time to obtain an estimate becomes unreasonable. Therefore, this method is tested no further than this number of samples. Computational costs will be discussed further in Section 3.12. Nevertheless, estimates for entropy, KL divergence, and now, also mutual information using this technique are quite accurate with a behavior similar to the one-dimensional case in Section 3.5. Numerical integration of the KDE density estimate further increases the computational cost with the procedure becoming unfeasible for a number of samples larger than 10,000. Further, integration only notably improves performance for mutual information.

Binning-based estimation does not suffer from the increase in computational cost as much as KDE-based estimation, but there is a small increase (see Section 3.12). As opposed to the one-dimensional cases investigated so far, binning yields large relative errors when estimating entropy even for larger sample sizes. The main drawback of the different binning rules is their inability to account for the multiple dimensions of the data, making the method less effective for dimensions higher than one. Here, Sturges’ rule is superior to the other methods, as it yields relatively accurate estimates for entropy and mutual information for samples of size 1000 or larger. For KL divergence, Scott’s and FD’s rules seem more effective. It appears that, for entropy and mutual information, a smoother histogram created using a larger Δ is more favorable, while for KL divergence, a more detailed histogram with a finer Δ is preferred. Recall that the QS approach has not been extended to higher dimensions yet and, hence, cannot be analyzed in this and the following multi-dimensional test cases.

Finally, for *k*-NN estimation, the findings from the previous one-dimensional cases seem to hold, at least for two dimensions. Namely, estimation performs best for smaller values of *k* both for entropy and KL divergence. This is not true for mutual information, where a large value for k=15 gives better estimates across all sample sizes. Importantly, for certain cases, particularly in Figure 10 shown for small sample sizes and k=1, the second term in Equation (Equation 29) becomes larger than all other terms, resulting in an estimation of negative mutual information. As mutual information cannot be negative per its definition (Equation (Equation 7)), the estimator reports 0.0 as the result. Hence, the *k*-NN estimate converges to the true mutual information systematically from below.

### 3.8. Case 5: 2D, Normal Mixture Distribution

To increase the level of complexity while exploiting the fact that two dimensions still allow for visualization, we again tested on a mixture of now bivariate normal distributions using the scheme in Equation (Equation 37).

#### 3.8.1. Numerical Reference Solution

As in Case 3, analytical expressions for entropy, KL divergence, or mutual information do not exist for this type of distribution. We, therefore, relied on numerical integration to obtain the reference values presented in the results. Figure 11a shows an equally weighted mixture of $N_{2}\left(\mu_{0},\;\Sigma_{0}\right)$ and $N_{2}\left(\mu_{1},\;\Sigma_{1}\right)$.

The parameters of the mixture are:$$\begin{array}{cc} \mu_{0}={\left[\begin{array}{cc} -2 & 0 \end{array}\right]}^{⊺} & \mu_{1}={\left[\begin{array}{cc} 2 & 0 \end{array}\right]}^{⊺}\\ \; & \\ \Sigma_{0}=\left[\begin{array}{cc} 1 & -0.5\\ -0.5 & 1 \end{array}\right] & \Sigma_{1}=\left[\begin{array}{cc} 1 & 0.5\\ 0.5 & 1 \end{array}\right] \end{array}$$

Figure 11b shows an approximating bivariate normal distribution $N_{2}\left(\mu_{2},\;\Sigma_{2}\right)$. The parameters of this distribution come from making the incorrect assumption that the $x_{0}$, $x_{1}$ pairs shown in Figure 11a are independent.
$$\begin{array}{cc} \mu_{2}={\left[\begin{array}{cc} 0 & 0 \end{array}\right]}^{⊺} & \Sigma_{2}=\left[\begin{array}{cc} 5 & 0\\ 0 & 1 \end{array}\right] \end{array}$$

#### 3.8.2. Estimation Results

The results of this test case are found to be very similar to the previous case of a bivariate normal distribution, with the most notable difference being that KDE seems to struggle more when moving away from a pure normal distribution, meaning that the performance is overall worse in this case.

Note that, for estimating mutual information with the *k*-NN method, further tests were conducted using k=50, but there were no noticeable differences between this value of *k* and the maximum shown in Figure 12 of k=15.

### 3.9. Case 6: 2D, Gamma-Exponential Distribution

An example of a skewed distribution in two dimensions is the gamma-exponential distribution, which has the following PDF [21], defined for $x_{1},x_{2}>0$:(42)$$p\left(x_{1},x_{2}\right)=\frac{x_{1}^{\theta }e^{-x_{1}-x_{1}\cdot x_{2}}}{\Gamma \left(\theta \right)}$$
where θ is the scaling parameter of the distribution >0 and Γ is the gamma function.

#### 3.9.1. Analytical and Numerical Reference Solutions

Darbellay and Vajda [45] presented a list of multivariate differential entropies and mutual information. According to them, the (joint) entropy of the gamma-exponential distribution is:(43)$$H\left(X_{1},X_{2}\right)=1+\theta -\theta \cdot \;\psi \left(\theta \right)+\ln\Gamma \left(\theta \right)-\ln\left(1\right)$$
where ψ is the digamma function or $\frac{d}{d\theta }\ln\Gamma \left(\theta \right)$. The mutual information is:(44)$$I\left(X_{1},X_{2}\right)=\psi \left(\theta \right)-\ln\left(\theta \right)+\frac{1}{\theta }$$

For calculating entropy and mutual information in this case, the scaling parameter was set to θ=3. For determining KL divergence, the parameter of the approximating distribution was set to θ=4. Because no reference in the literature was found to provide a theoretical result of relative entropy between two gamma-exponential distributions, numerical integration was used to obtain the true value of $D_{KL}$ presented in the results. Figure 13 shows a plot of the two PDFs used in this test case.

#### 3.9.2. Estimation Results

The estimation results for the case of the gamma-exponential distribution are shown in Figure 14. Here, we have the case of a distribution that has a different shape than the normal distribution; therefore, the limitations of the usage of a Gaussian kernel in the KDE-based estimator become more apparent. The KDE-based entropy estimate only achieves an accurate result for large sample sizes and using numerical integration. This seems to contradict the previous cases, but this can be explained by how numerical integration is implemented on the KDE estimator. Numerical integration requires a set of limits, and as implemented, the limits were chosen as the maximum and minimum values available in the sample for each dimension, plus and minus the specific bandwidth calculated for the Gaussian kernel. Therefore, these limits constrain the region where numerical integration happens, even though the kernel-based representation of the distribution has support everywhere due to the usage of the Gaussian kernel. This effect also applies to KL divergence and mutual information, but is more apparent in the latter. Both of these estimates remain biased to a significant degree even with the largest sample sizes tested here.

Binning estimates applying Scott’s rule give the best results, while the FD rule drastically underestimates entropy and overestimates mutual information. Sturges’ rule shows the opposite behavior. As previously commented on, this is due to the FD rule calculating a smaller Δ and forcing a greater number of bins than Sturges’. This overestimation on the required number of bins causes the representation of the joint distribution p(x,y) to be sparser and to have a larger number of empty bins, leading to a smaller variability in the representation of p(x,y). Generally, estimating entropy with bins using Scott’s rule leads to small relative errors for practically all considered sample sizes, whereas the mutual information estimate using this same rule still contains quite high relative errors even for the largest sample size of 100,000. Because KL divergence compares the ratios of p(x) and q(x), very similar results are obtained across the binning hyperparameters as the absolute bin size does not matter as much as the fact that equal binning schemes are defined for p(x) and q(x).

As with the other cases, *k*-NN-based estimates improve in accuracy with increasing sample size, and k=1 performs best for entropy and KL divergence. In the case of entropy, larger *k* leads to a worse approximation, while still, all investigated values of *k* produce very small errors with large sample sizes. For KL divergence, however, the results become more mixed with larger sample sizes, where the highest considered *k* of 15 approaches the highly accurate result of k=1, while the smaller *k* values in between seem to stabilize at large relative errors, underestimating the true KL divergence significantly. In the case of mutual information, larger *k* is favorable for all sample sizes, but all *k* larger than one achieve acceptable relative errors for sample sizes larger than 1000.

### 3.10. Case 7: 4D, Multivariate Normal Distribution

For higher dimensions, we tested the estimators’ ability to quantify the entropy, KL divergence, and mutual information of a four-dimensional normal distribution, since analytical solutions are available. We adopted the experiments by Wang et al. [23] and replicated their results, while going further with their experiment by quantifying the sampling uncertainty of their results, as well as investigating additional estimation methods. For entropy and mutual information, data were sampled from $N_{4}\left(\mu_{0},\;\Sigma_{0}\right)$, and for KL divergence, data were also sampled from $N_{4}\left(\mu_{1},\;\Sigma_{1}\right)$.
$$\begin{array}{cc} \mu_{0}={\left[\begin{array}{cccc} 0.1 & 0.3 & 0.6 & 0.9 \end{array}\right]}^{⊺} & \mu_{1}={\left[\begin{array}{cccc} 0 & 0 & 0 & 0 \end{array}\right]}^{⊺}\\ \; & \\ \Sigma_{0}=\left[\begin{array}{cccc} 1 & 0.5 & 0.5 & 0.5\\ 0.5 & 1 & 0.5 & 0.5\\ 0.5 & 0.5 & 1 & 0.5\\ 0.5 & 0.5 & 0.5 & 1 \end{array}\right] & \Sigma_{1}=\left[\begin{array}{cccc} 1 & 0.1 & 0.1 & 0.1\\ 0.1 & 1 & 0.1 & 0.1\\ 0.1 & 0.1 & 1 & 0.1\\ 0.1 & 0.1 & 0.1 & 1 \end{array}\right] \end{array}$$

#### 3.10.1. Analytical Reference Solution

In this case, we can use Equations (Equation 39) and (Equation 40) as analytical expressions for entropy and KL divergence, respectively.

For mutual information, Arellano-Valle et al. [46] suggested breaking down the distribution into marginal distributions with *n* and *m* dimensions, where d=n+m and:(45)XY∼Nn+mμXμY,ΣXXΣXYΣYXΣYY

In this form, using the expression for mutual information in the first row of Equation (Equation 7) and considering the expression for entropy obtained in Equation (Equation 39), mutual information can be written as:(46)$$I\left(X,Y\right)=\frac{1}{2}\ln\left(\frac{\det\left(\Sigma_{XX}\right)\det\left(\Sigma_{YY}\right)}{\det\left(\Sigma \right)}\right)=-\frac{1}{2}\ln\left(\det\left(I_{n}-\Sigma_{XX}^{-1}\Sigma_{XY}\cdot \Sigma_{YY}^{-1}\Sigma_{YX}\right)\right)$$

As we cannot present plots of the PDF of the distributions used in this application case, we show here the reference true values:
$$H(X)=5.09\;\mathrm{nats}\;D_{KL}\left(p||q\right)=0.90\;\mathrm{nats}\;I(X;Y)=0.24\;\mathrm{nats}$$

The reference solution for mutual information comes from applying Equation (Equation 45) to the distribution $N_{4}\left(\mu_{0},\;\Sigma_{0}\right)$. The distribution is split so that the distribution of *X* contains the first three dimensions of the original distribution; therefore n=3; the distribution of *Y* has dimensions of the highest order (d=4) and m=1. More succinctly, mutual information is calculated as: $I(\left[x_{1},x_{2},x_{3}\right];x_{4})$.

#### 3.10.2. Estimation Results

The estimation results for the case of the 4D normal distribution are shown in Figure 15. In higher dimensions than those of the previous cases, the main drawbacks of some of the estimation methods start to become apparent. To begin, numerical integration of a KDE-based probability distribution becomes too computationally expensive, so this method was not considered in this experiment. Further, as was also the case for some of the previous cases, the evaluation of a KDE-based representation also becomes too expensive for larger sample sizes; therefore, the KDE-based direct resubstitution estimate was only calculated up to 25,000 samples. For entropy and mutual information, it seems that the estimator would converge for a larger number of samples, with the estimates having very little variance after 1000 samples. The estimate for KL divergence, however, is extremely stable with a highly biased result (40% relative error).

Binning-based estimation performs acceptably well only when using Sturges’ rule. Sturges’ rule typically results in the largest bin width, with Scott’s and FD’s rules giving smaller bin widths and, therefore, a larger amount of bins in the histogram-based representation of the PDF. In the case of entropy, for smaller bins, the contribution of each bin to the computation of entropy is very small and the correction factor described in Section 2.2 and Equation (Equation 20) dominates the calculation, typically underestimating entropy and resulting in values that are not shown in the limits of the plot in Figure 15. Only Sturges’ rule with its larger bin sizes converges to the true value with the highest investigated sample size, however, approaching the true value very steeply, meaning that all smaller sample sizes exhibit large relative errors. The same is true for mutual information, but this quantity is usually overestimated with smaller sample sizes. Finally, also for KL divergence, Sturges’ is the only rule that shows a reasonable convergence behavior. The other rules result in many more bins, making it difficult for samples to match the same bin as those that are evaluated for the computation of KL divergence between the histogram of $\hat{p}\left(x\right)$ and $\hat{q}\left(x\right)$. Contrary to the previous case, in higher dimensions, not only equal binning schemes matter, but also the quality of the estimate depends on the size of the bin. As the data are sparser in higher dimensions, samples that are close together should be accounted for in the same bin, and this becomes difficult when the number of bins is too large.

*k*-NN-based estimation performs best among all investigated methods and quantities with similar behavior as in the previous lower-dimensional cases, where, again, k=1 is the best hyperparameter for entropy and KL divergence, while k=15 seems best for mutual information. For mutual information, it can also be noted that k=1 is not an adequate parameter for estimation, especially for small sample sizes, as the estimator typically reports a value of 0.0.

### 3.11. Case 8: 10D, Multivariate Normal Distribution

Following the experiments by Wang et al. [23] and, therefore, using a similar notation as the previous case, for entropy and mutual information, data were sampled from $N_{10}\left(\mu_{0},\;\Sigma_{0}\right)$, and for KL divergence, data were sampled from $N_{10}\left(\mu_{1},\;\Sigma_{1}\right)$. In this case, both distributions are centered at 0, and we write the covariance matrices using the notation *i* for rows and *j* for columns, then $\Sigma_{0}^{i,i}=1$, $\Sigma_{0}^{i,j}=0.9$ and $\Sigma_{1}^{i,i}=1$, $\Sigma_{1}^{i,j}=0.1$, for iandj=1,…,10.

#### 3.11.1. Analytical Reference Solution

We can again use Equations (Equation 39), (Equation 40), and (Equation 46). Similar to the previous case, to calculate mutual information, Equation (Equation 45) is applied to the distribution $N_{10}\left(\mu_{0},\;\Sigma_{0}\right)$, and once again, the dimensions are separated so that n=9 and m=1, where *m* is the dimension of the highest order; then, mutual information is calculated as $I(\left[x_{1},\ldots ,x_{9}\right];x_{10})$. The following solutions are obtained as reference values:
$$H(X)=4.93\;\mathrm{nats}\;D_{KL}\left(p||q\right)=7.00\;\mathrm{nats}\;I(X;Y)=1.10\;\mathrm{nats}$$

#### 3.11.2. Estimation Results

KDE-based estimation performs poorly for all quantities. For estimating entropy, the kernel-based representation of the PDF seems to not be as smooth as the true distribution, resulting in a lower estimation of entropy. Nevertheless, the representation does improve with additional samples, and it would be expected that, given enough samples, the KDE PDF would resemble the true distribution and the estimator would converge. Then, again, more samples would mean added computational cost in evaluating the kernel representation of the PDF, and this procedure becomes a limiting factor. As with the previous case, the maximum number of samples for entropy and KL divergence has been limited to 25,000. For KL divergence, the mean estimates across sample sizes are very consistent, but the kernel-based representation of both the true p(x) and approximating q(x) distributions do not resemble the distributions from which the data were sampled; therefore, a biased result is obtained. Finally, mutual information is consistently overestimated across all sample sizes, with added computational cost, as three kernel-based approximations of the true distribution have to be evaluated: p(x,y), p(x), and p(y).

Considering binning methods, similar to the previous case, only Sturges’ rule is able to capture an adequate representation of the underlying distribution with its larger bin widths. For entropy, as the sample size increases, the estimate made by Sturges moves toward the true value. But, even for the largest sample size of 100,000, the estimate made using this rule underestimates the true value of entropy by approximately 20%. Furthermore, when using rules with smaller bin widths, bin occupations and, hence, densities become even lower, resulting in the estimated entropy being typically negative. This can be seen in Figure 16 as the trajectory that Scott’s rule follows, and this was true also for the previous case in Section 3.10. Further, for KL divergence, not even the large bins produced by Sturges’ rule prevent the mismatch between the bin occupations for high-dimensional data, making the result of estimating KL divergence equivalent to infinity and not shown in Figure 16. Finally, in the case of mutual information, once again, only Sturges’ rule produces an estimate that follows the expected behavior of improving as the sample size is increased. Both Scott’s and FD’s rule produce more bins, which typically have higher estimated densities in the joint distribution of *X* and *Y* than the product of both marginal distributions *X* and *Y*. As mutual information measures the distance between a joint distribution and the product of its marginals, as described in Equation (Equation 6), mutual information increases with the sample size, as can be seen in Figure 16, and said increase is not compensated by dividing by the size of the sample, as Equation (Equation 22) indicates.

*k*-NN-based estimation continues to perform well, but the challenges of estimation in higher dimensions also affect the results based on this method. Here, the number of samples was also limited from the typical 100,000 as the maximum evaluated to 50,000. Although not as markedly as with other methods, the computational cost increased from seconds to minutes in comparison to the previous four-dimensional case, even considering that the number of available samples was restricted. Notably, the estimator gives a much more accurate result for entropy across all sample sizes than KDE or binning for the largest implemented sample size, using the *k*-NN hyperparameter k=1. For KL divergence and mutual information, the performance is not as accurate, even when considering an optimal choice of hyperparameters. For KL divergence, estimates using k=1 show a downward trend, which would suggest that the estimator will eventually converge, but at a much higher number of available samples. The same is true for mutual information using k=15. However, the relative errors of both quantities are much smaller than those produced by KDE or binning and remarkably similar in order of magnitude to all lower-dimensional test cases.

### 3.12. Computational Cost

A desirable property of an estimator is a low computational time. With such a property, the estimator would lend itself to practical purposes and processes that require a result to be estimated many times such as Monte Carlo methods or bootstrapping [10].

To assess the computational cost of the different estimation methods, we chose to evaluate the computation time required for each test case for a set number of samples and one hyperparameter of each specific method. Given the results presented above, we chose to use 10,000 samples to evaluate computation time because, in most cases, at this sample size, each estimator has (almost) converged to the true result. In terms of hyperparameters, for the binned estimator, we chose to use Scott’s method to calculate an appropriate bin width (see Section 2.2), Silverman’s bandwidth for the KDE estimator (Equation (Equation 14)), and k=1 for the *k*-NN estimator to calculate entropy and KL divergence, while for mutual information, we chose k=15. The results in Table 2 show the mean, maximum, and minimum computation time (in seconds) for each experiment. All results were obtained on a single thread of an Intel(R) Xeon(R) CPU E5-26280 v2 with a clock speed of 2.80 GHz.

Across the board, Table 2 shows that binning- and *k*-NN-based methods have the lowest required computational time with a single estimate taking less than a second in most cases. This makes them ideal for tasks that require repeating calculations for different samples of data or using different hyperparameters.

For the binning estimator, the process of creating, filling, and applying a specific equation as described in Section 2.2.1 typically takes milliseconds with an increase depending on the number of dimensions of the data. While, for lower dimensions, the required histogram is calculated, for the 4- and 10-dimensional cases, we employed a procedure in which every point in the sample is replaced by the specific bin it occupies, greatly limiting the amount of memory and computational time required for estimation. Interestingly, there is a decrease in the time required to estimate KL divergence between the 4- and 10-dimensional cases. This is the product of having to find the matching and occupied bins given by Scott’s rule-of-thumb for estimating bin size. As the bins are smaller, they are more sparsely populated in the 10-dimensional case, resulting in less matching bins that are also occupied. Having to calculate the estimate of KL divergence for less bins makes this process faster in the 10-dimensional case when compared against the 4-dimensional case.

As introduced by Gupta et al. [10], the QS estimator uses bootstrapping twice. From a single sample, it uses bootstrapping to determine the theoretical quantiles that best approximate the true distribution, and then, it uses bootstrapping again to estimate the bootstrapped confidence intervals of the estimate. Because the experiments wanted to address the effect of sampling variability on the estimators, the second step of the QS estimator was not performed and bootstrapping was only performed to determine the quantiles. This is reflected in the computational times shown in Table 2 as the QS estimator takes longer than histogram-based methods using rules-of-thumb to determine the bin width. As described in Section 2.2.2, bootstrapping was performed $N_{k}=500$ times to determine the ideal quantiles.

KDE-based estimation provides the estimates with the highest computation time out of all the estimation methods evaluated. Calculating a direct resubstitution estimate for all quantities is one or two orders of magnitude slower than using any of the previously discussed methods. This becomes even worse when numerical integration is used to calculate an estimate. For a resubstitution estimate, each data point in the sample has to be evaluated once in the KDE-based representation of the PDF. This is opposed to performing numerical integration where the number of evaluations is unknown and they continue until a certain tolerance for error in the method is met. This is apparent as, for lower-dimensional cases, the number of required evaluations for the numerical integration estimate is low, making this method faster than evaluating all available samples in the kernel-based PDF for a resubstitution estimate. Nevertheless, due to its high computation time, the KDE-based estimator does not seem suitable for multiple evaluations.

Finally, *k*-NN estimation requires about 10-times the computational effort of binning in low dimensions, but it’s faster than direct KDE evaluation by about a factor of 1000. In high-dimensional cases, the effort of *k*-NN increases, but still is well below the effort of the direct evaluation of KDE.

### 3.13. Synthesis of Findings from Test Cases

Histogram-based estimation or binning is the most common approach for calculating entropy, KL divergence, or mutual information. In this study, we found that it is an accurate method especially for data in one dimension and using Scott’s rule-of-thumb to determine an adequate binning scheme for the data. Nevertheless, the method loses accuracy for data in higher dimensions, where there are no established good practices for selecting a particular bin size, and the rules-of-thumb extrapolate poorly to higher dimensions. Only Sturges’ rule, which calculates the largest bin widths, seems to generalize to higher dimensions, as the available data for estimation are sparser and less bins are able to capture an adequate number of points in a particular bin.

As an alternative, KDE-based estimation uses a kernel to create a smooth representation of a probability density function for each point in the available sample. The performance of this method is good when compared to the other estimation methods, but it might not be practical due to its high computational cost and the fact that, as implemented in this study, it used a Gaussian kernel and it was applied to data sampled from normal distributions (multivariate normal distributions in higher dimensions). Extrapolating from the lower-dimensional test cases, we reckon that, for high-dimensional distributions with different shapes, this estimation method might not perform as well.

Across all of the experiments, *k*-NN-based estimation was among the top-scoring methods, independent of the shape and dimensionality of the distribution sampled from, and the given sample size. We found a very consistent best-performing choice of the hyperparameter *k*, i.e., the number of neighbors to consider in the formulation of the estimator. Notably, the optimal choice of *k* depends on the specific quantity to be estimated: for entropy and KL divergence, k=1 performed best, while for mutual information, a high number of k=15 was shown to be optimal. Higher values of *k* were tested, but did not show significant improvement, while using k=1 for mutual information is generally not recommended. These findings suggest that a direct estimation in the immediate vicinity of each sample point is beneficial for estimating entropy (thereby, not smearing out the sample distribution too much) and KL divergence (where the ratios of probabilities are estimated per sampling point, thereby not accumulating approximation errors before calculating the ratios). For mutual information, a more aggregated view of the sample is required, favoring a higher number of neighbors to consider.

Moreover, the computational cost is relatively low, as described in Section 3.12. Hence, *k*-NN-based estimation is readily implementable in practical applications and lends itself to repeated sampling such as in Monte Carlo or bootstrapping analyses.

We found that KDE and binning both performed well in some cases, but by far not as consistently as *k*-NN and with varying methods (e.g., direct estimation vs. integration for KDE and Scott’s rule vs. Sturges’ and FD’s rule for binning). Hence, the clear recommendation on how to set the value of *k*-NN’s hyperparameter is a further advantage of this method.

## 4. Summary, Conclusions, and Outlook

Estimating information-theoretic quantities from sample data, possibly in higher dimensions, poses a challenge. Most methods rely on an initial density estimation step, but density estimation itself is known to be computationally demanding or even prohibitive, and to produce unreliable results with unknown accuracy and precision in practical settings. Further, most methods assume a certain shape of distribution that the sample stems from, with an unknown impact on the estimator’s performance if the true distribution deviates from that assumption. As an alternative, nearest-neighbor-based methods to directly estimate specific information-theoretic quantities have been proposed, skipping the initial density estimation step altogether. Hence, they show promise to also perform well for higher dimensions and arbitrary distributions, but have never been systematically compared to density-based methods such as kernel density estimation (KDE) or binning (histogram-based schemes). In fact, most users in science and practice seem to favor binning for its straightforward implementation, but with no further justification. We hypothesize that this is due to a lack of systematic evaluation and guidance on which method to choose in what settings. With this investigation, we aimed to close this research gap.

Typically, a new method was introduced individually without systematic comparative analysis in regard to other methods. In some cases, a brief comparison was made against another method that uses the same basic principle, such as histograms or *k*-NN. In response, we presented and discussed three of the most widely used non-parametric estimation methods for information-theoretic quantities, namely binning, KDE, and k-nearest neighbors (*k*-NN). To evaluate the estimators’ performance, we designed test cases that used data samples from distributions with different shapes and with different numbers of dimensions to quantify the information-theoretic quantities entropy, KL divergence, and mutual information. Depending on the level of complexity of each case, analytical solutions existed, or were approximated with a high-quality numerical reference solution. We tested the estimation methods on each case and reported the performance in a chart that serves for intercomparing all methods for all target quantities as a function of sample size. The true distributions used to generate the data for our experiments ranged from simple 1D uniform or normal distributions over more skewed or bimodal shapes up to multivariate distributions in 4 and 10 dimensions. Sample sizes ranged from 100 to 100,000, with a reduction of the maximum sample size in cases where computational effort exploded. As described, this is a very practical and easy-to-understand basis for our study. We also accounted for sampling variability as each experiment was repeated with 300 random seeds for the data-generating sampling procedure. Further, we considered different choices of hyperparameters for each estimation method. Finally, we assessed and compared the computation time required to obtain an estimate for a sample size of 10,000, which produced a well-converged estimate for most estimation methods. All methods showed larger relative errors for sample sizes typically below 1000, pointing to the fact that samples of such a small size are not representative enough of the underlying true distribution, and hence, any estimation method necessarily fails in reliably quantifying the underlying distribution’s particular property. For larger sample sizes, however, estimation results typically converged to a more or less biased estimate, with distinct differences between the methods.

For binning, the most important parameter is the bin width or Δ used to build the histogram because it controls the trade-off between oversmoothing and undersmoothing the data with respect to the true distribution. Typical rules-of-thumb were implemented and tested in this study (namely Scott’s, Sturges’, and FD’s rule), as well as the Quantile Spacing method by Gupta et al. [10], but alternative methods using piecewise constant approximations for density [47], as well as methods that calculate an estimate of Δ based on the minimization of the mean-integrated-squared error (MISE) should be explored as well in the future. These latter methods aim to reduce the error between the true distribution f(x) and the histogram-based distribution $\hat{f}\left(x|\Delta \right)$ [48].

Our KDE-based estimator used a Gaussian kernel as a typical choice; however, the choice of kernel is a hyperparameter that was not analyzed in this study, and it is expected that the performance of the estimator would vary depending on the choice of kernel used. A boxcar kernel, for example, would improve estimation in cases where the true distribution has distinct bounds such as a uniform distribution in one dimension, or a gamma-exponential distribution in two dimensions. However, such improvement would be accompanied by the known high computational cost of kernel-based methods, and as such, if only the particular information-theoretic quantity is of interest and no other property related to the kernel-based representation of the PDF, other more efficient methods seem better suited.

Finally, the evaluation of the *k*-NN-based estimator proved its performance and served as a showcase for its computational efficiency. The speed of the estimator comes from the usage of the *k*-d Tree data structure, which is in charge of the look-up operation required to identify nearest neighbors. While we found that, in its original form, the estimator performs well in the diverse set of experiments we tested it on, recent variations have been proposed to further improve the estimation of entropy by combining *k*-NN with normalizing flows [49] and the estimation of mutual information by combining *k*-NN with KDE [22] or using neural networks [50].

In general, across all test cases, *k*-NN produced among the lowest relative errors in estimating entropy, KL divergence, and mutual information, independent of the sample size and shape and dimensionality of the distribution sampled from. Also, a clear identification of the best-performing value of the hyperparameter *k*, i.e., the number of neighbors to be considered in the vicinity of a sampling point, was possible; interestingly, the optimal choice of *k* depends on the specific quantity to be estimated. For entropy and KL divergence, k=1 performed best, while for mutual information, a more integral approximation with a high number of k=15 was shown to be optimal (even higher values were tested, but did not show significant further improvement).

All tested methods might benefit from a specifically tailored bootstrapping approach to increase the stability of the estimator for large samples that are (sufficiently) representative of the true underlying distribution; this is recommended for further analysis. For small sample sizes, however, our study has confirmed the dominant impact of non-representativeness.

Based on our evaluation of the performance, ease of implementation, and computational effort, we recommend *k*-NN-based estimation for estimating information-theoretic quantities from sample data, especially in higher dimensions, due to its clear advantages. However, the significance of this work extends beyond a theoretical comparison and practical performance assessment: we have also collected these methods in a publicly available Python 3 toolbox, ensuring transparency and accessibility for the wider research community. This toolbox shall serve as a valuable resource, enabling researchers and practitioners to integrate information-theoretic concepts into their data analysis and modeling workflows. Specifically, this study in combination with the toolbox shall enable them to make a well-informed decision on the choice of estimation method.

## Figures and Tables

**Figure 1 entropy-26-00387-f001:**
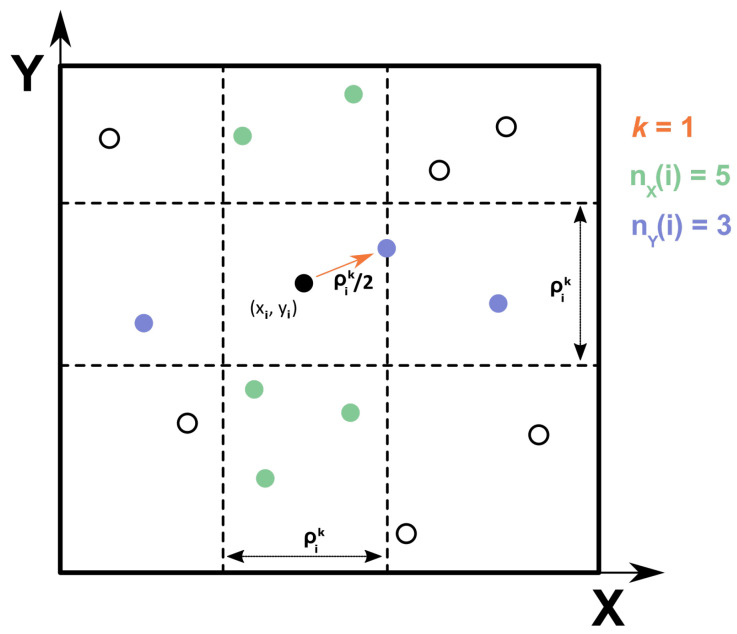
Determining $\rho_{i}^{k}$, $n_{x}\left(i\right)$, and $n_{y}\left(i\right)$ for a single pair $\left(x_{i},y_{i}\right)$ in the algorithm for Equation (Equation 29). Adapted from Kraskov et al. [21].

**Figure 2 entropy-26-00387-f002:**
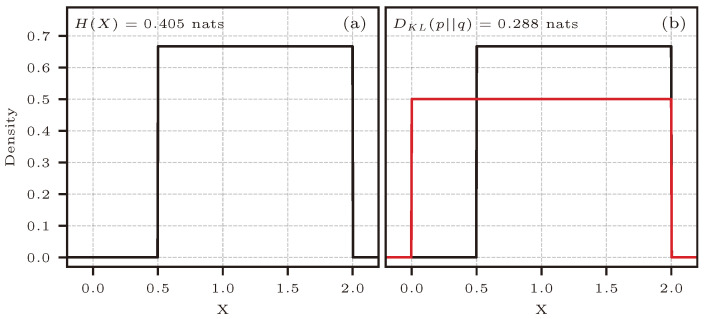
Case 1: 1D, uniform distribution. (**a**) PDF of a uniform distribution with a,b=0.5,2.0 and a reference value for entropy *H*; (**b**) same as (**a**) including the PDF of an approximating uniform distribution (in red) with c,d=0.0,2.0 and a reference value for KL divergence $D_{KL}$.

**Figure 3 entropy-26-00387-f003:**
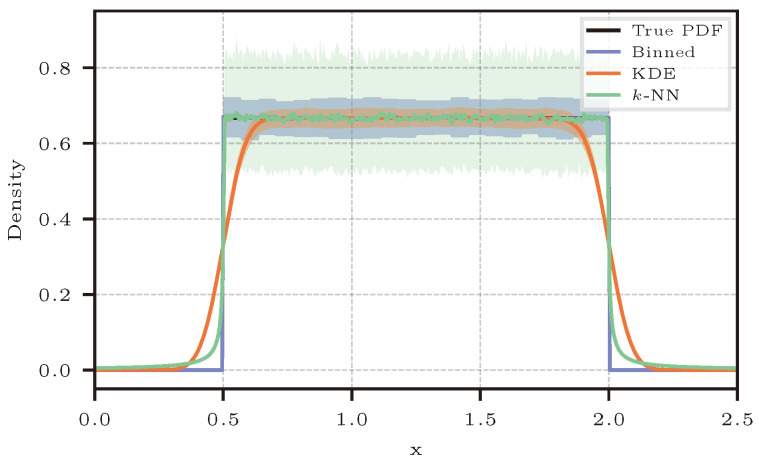
Comparison of all estimation methods for density estimation in Case 1 (1D, uniform distribution).

**Figure 4 entropy-26-00387-f004:**
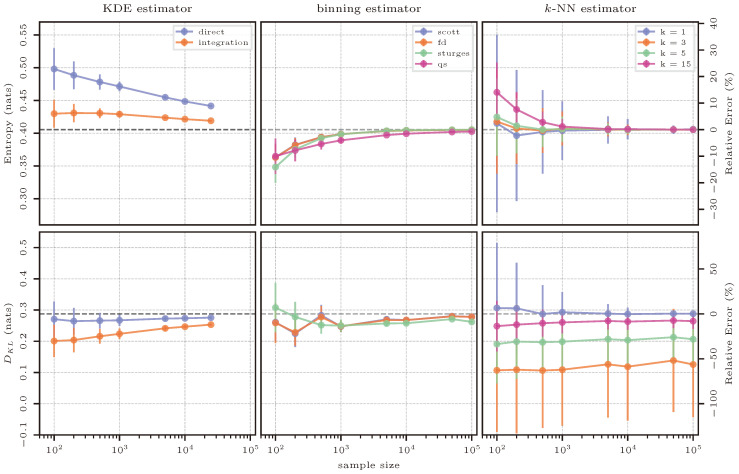
Evaluation of all estimation methods (columns) for entropy (**top**) and KL divergence (**bottom**), in Case 1 (1D, uniform distribution).

**Figure 5 entropy-26-00387-f005:**
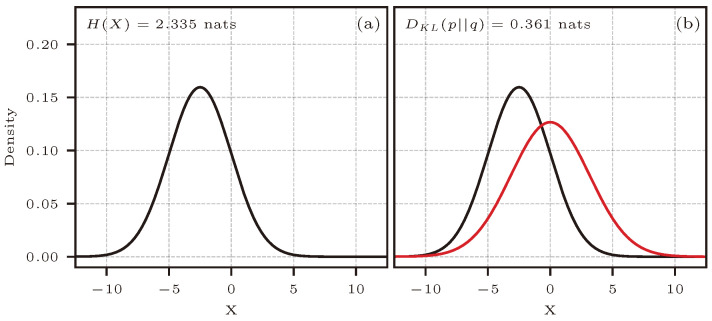
Case 2: 1D, normal distribution. (**a**) PDF of a normal distribution $N(-2.5,\;2.5^{2})$ and a reference value for entropy *H*; (**b**) same as (**a**) including the PDF of an approximating normal distribution (in red) $N(0,\;3.15^{2})$ and a reference value for KL divergence $D_{KL}$.

**Figure 6 entropy-26-00387-f006:**
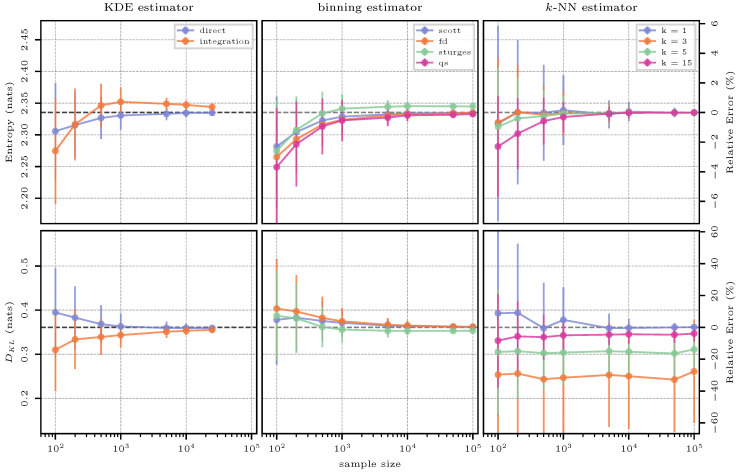
Evaluation of all estimation methods (columns) for entropy (**top**) and KL divergence (**bottom**), in Case 2 (1D, normal distribution).

**Figure 7 entropy-26-00387-f007:**
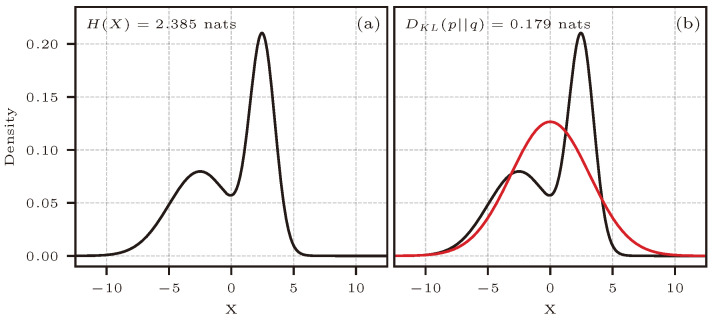
Case 3: 1D, normal mixture distribution. (**a**) PDF of a normal mixture distribution (see the parameters in the text) and a reference value for entropy *H*; (**b**) same as (**a**) including the PDF of an approximating normal distribution (in red) $N(0,\;3.15^{2})$ and a reference value for KL divergence $D_{KL}$.

**Figure 8 entropy-26-00387-f008:**
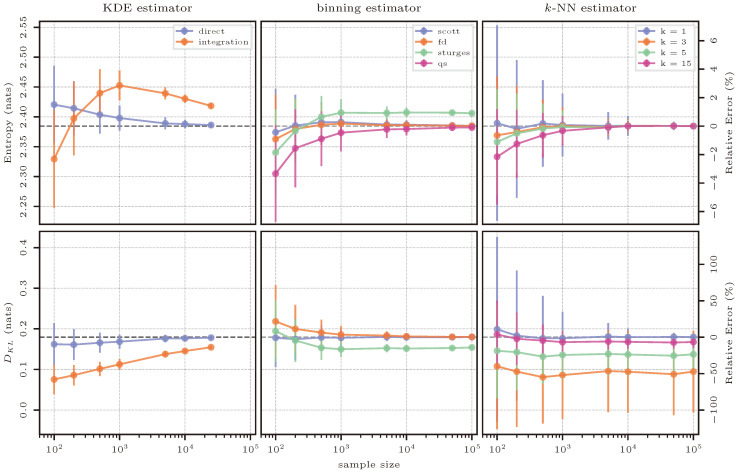
Evaluation of all estimation methods (columns) for entropy (**top**) and KL divergence (**bottom**), in Case 3 (1D, normal mixture distribution).

**Figure 9 entropy-26-00387-f009:**
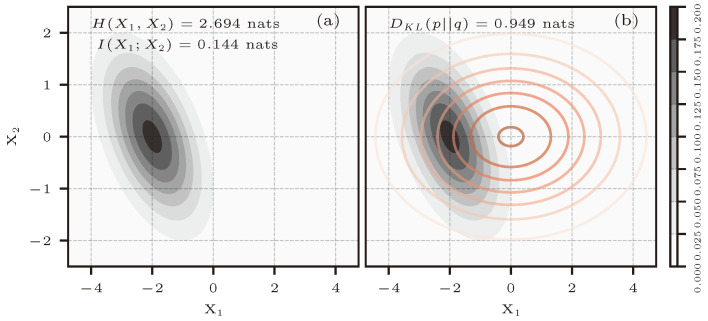
Case 4: 2D, normal distribution. (**a**) PDF of a bivariate normal distribution (see the parameters in the text) and reference values for entropy *H* and mutual information *I*; (**b**) same as (**a**) including the PDF of an approximating bivariate normal distribution (in red; see the parameters in the text) and a reference value for KL divergence $D_{KL}$.

**Figure 10 entropy-26-00387-f010:**
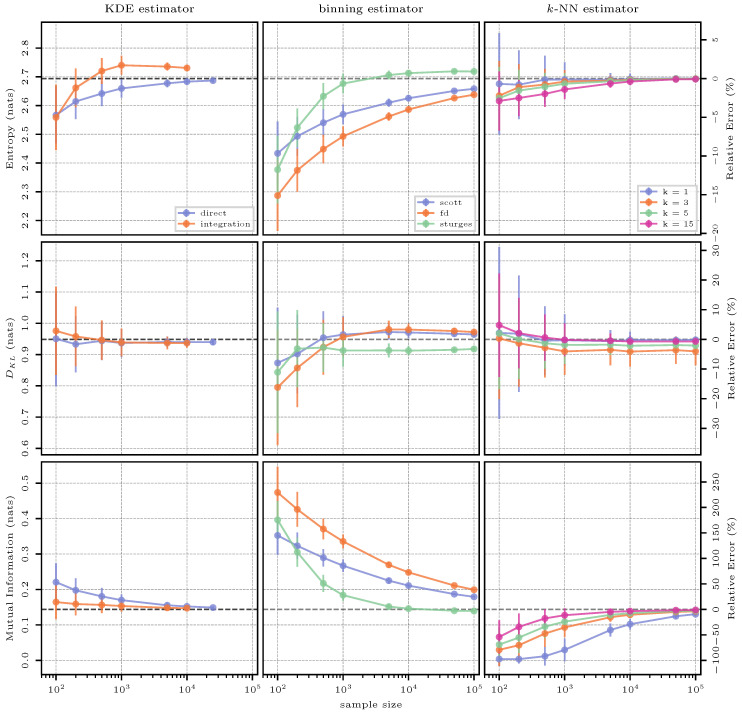
Evaluation of all estimation methods (columns) for entropy (**top**), KL divergence (**middle**), and mutual information (**bottom**), in Case 4 (2D, normal distribution).

**Figure 11 entropy-26-00387-f011:**
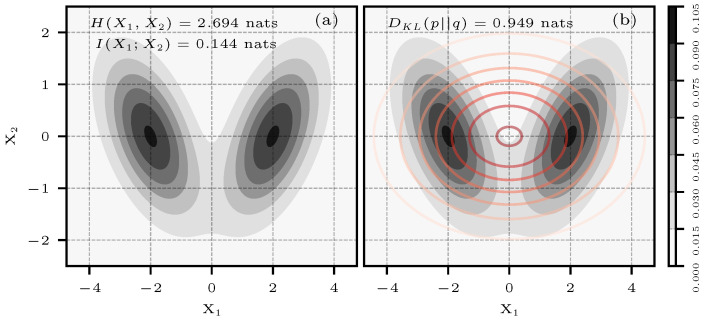
Case 5: 2D, normal mixture distribution. (**a**) PDF of a bivariate normal mixture distribution (see the parameters in the text) and reference values for entropy *H* and mutual information *I*; (**b**) same as (**a**) including the PDF of an approximating bivariate normal distribution (in red; see the parameters in the text) and a reference value for KL divergence $D_{KL}$.

**Figure 12 entropy-26-00387-f012:**
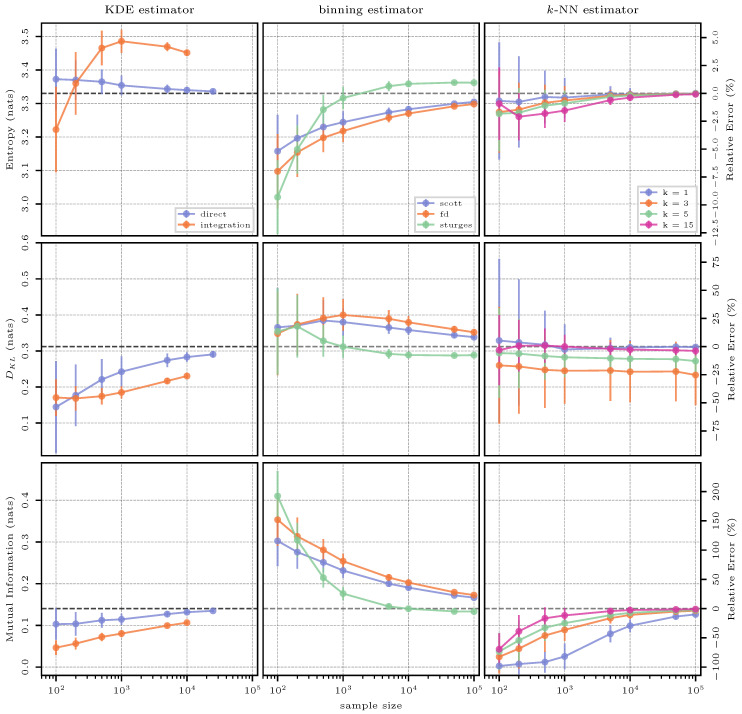
Evaluation of all estimation methods (columns) for entropy (**top**), KL divergence (**middle**), and mutual information (**bottom**), in Case 5 (2D, normal mixture distribution).

**Figure 13 entropy-26-00387-f013:**
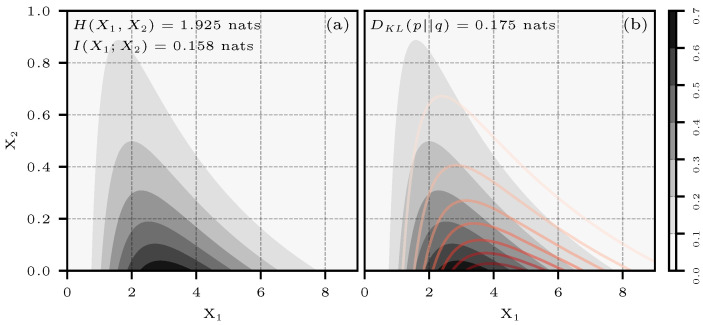
Case 6: 2D, gamma-exponential distribution. (**a**) PDF of the gamma-exponential distribution, where θ=3, and reference values for entropy *H* and mutual information *I*; (**b**) same as (**a**) including the PDF of an approximating function (in red), where θ=4, and a reference value for KL divergence $D_{KL}$.

**Figure 14 entropy-26-00387-f014:**
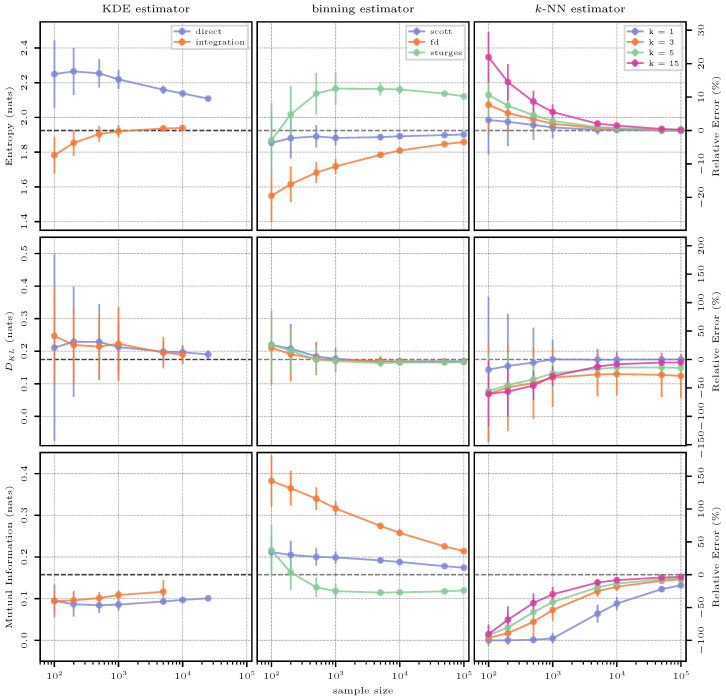
Evaluation of all estimation methods (columns) for entropy (**top**), KL divergence (**middle**), and mutual information (**bottom**), in Case 6 (2D, gamma-exponential distribution).

**Figure 15 entropy-26-00387-f015:**
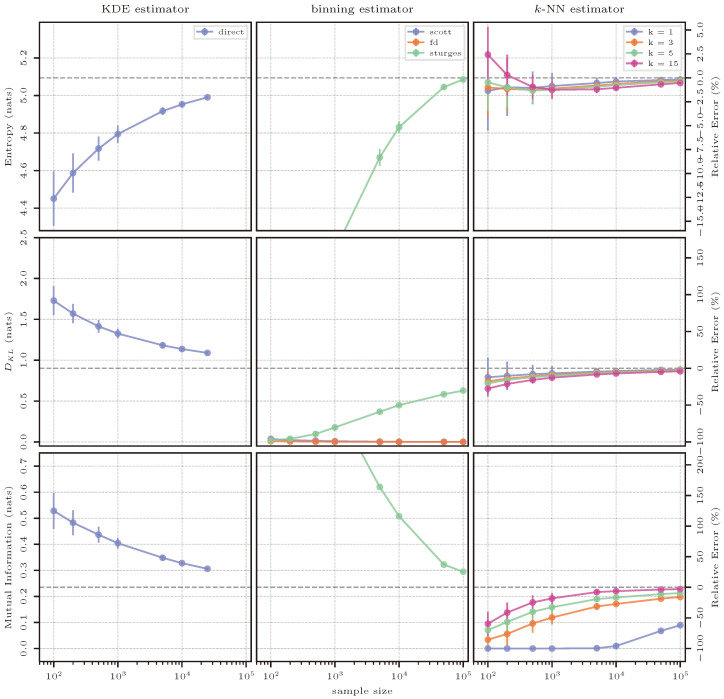
Evaluation of all estimation methods (columns) for entropy (**top**), KL divergence (**middle**), and mutual information (**bottom**), in Case 7 (4D, normal distribution).

**Figure 16 entropy-26-00387-f016:**
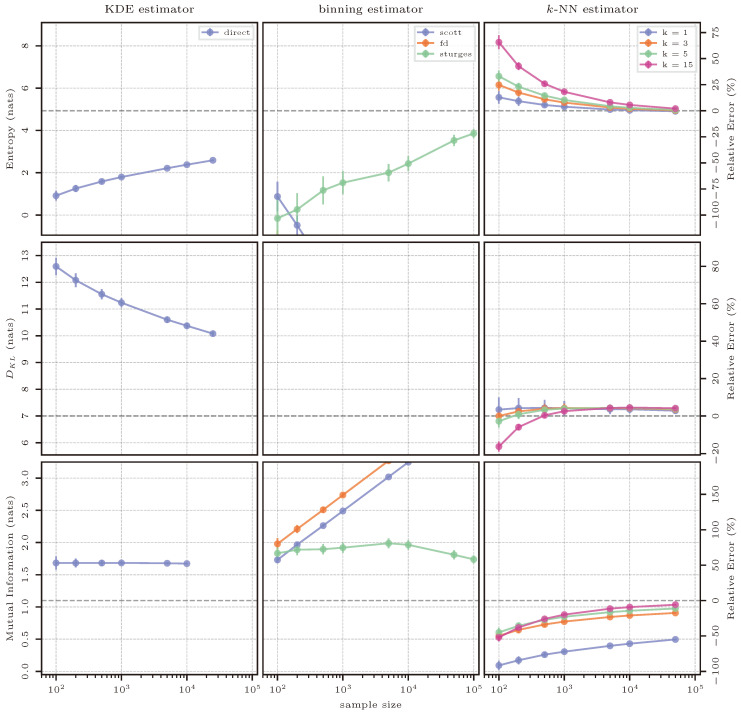
Evaluation of all estimation methods (columns) for entropy (**top**), KL divergence (**middle**), and mutual information (**bottom**), in Case 8 (10D, normal distribution).

**Table 1 entropy-26-00387-t001:** Summary of test cases.

Case ID	Distribution	No. of Dimensions
1	Uniform	1
2	Normal	1
3	Normal mixture	1
4	Bivariate normal	2
5	Bivariate normal mixture	2
6	Gamma-exponential	2
7	Multivariate normal	4
8	Multivariate normal	10

**Table 2 entropy-26-00387-t002:** Comparison of computational time between cases and methods (time in seconds).

Quantity	Case	Bins		KDE		*k*-NN
		Scott	QS	Direct	Integration	
Entropy	Uniform	$0.001_{0.001}^{0.012}$	$0.263_{0.137}^{0.556}$	$3.408_{1.643}^{4.078}$	$0.122_{0.091}^{0.290}$	$0.018_{0.014}^{0.046}$
	Normal	$0.001_{0.001}^{0.014}$	$0.278_{0.210}^{0.455}$	$3.467_{2.652}^{4.058}$	$0.286_{0.197}^{0.672}$	$0.018_{0.014}^{0.043}$
	Normal Mixture	$0.001_{0.001}^{0.001}$	$0.285_{0.225}^{0.505}$	$3.486_{2.795}^{4.196}$	$0.245_{0.156}^{0.604}$	$0.018_{0.014}^{0.040}$
	2D Normal	$0.003_{0.002}^{0.012}$	-	$3.513_{2.861}^{4.176}$	$80.09_{56.75}^{112.1}$	$0.023_{0.017}^{0.077}$
	2D Normal Mixture	$0.002_{0.002}^{0.014}$	-	$3.525_{2.667}^{4.048}$	$37.23_{21.06}^{49.74}$	$0.022_{0.018}^{0.056}$
	Gamma-Exponential	$0.003_{0.002}^{0.005}$	-	$3.548_{3.338}^{4.012}$	$279.2_{76.25}^{633.6}$	$0.023_{0.012}^{0.064}$
	4D Normal	$0.008_{0.006}^{0.015}$	-	$4.052_{2.825}^{4.896}$	-	$0.043_{0.034}^{0.115}$
	10D Normal	$0.016_{0.010}^{0.039}$	-	$5.193_{4.120}^{6.357}$	-	$0.597_{0.442}^{0.937}$
$D_{KL}$	Uniform	$0.002_{0.001}^{0.011}$	-	$4.712_{4.668}^{4.755}$	$0.524_{0.365}^{0.952}$	$0.034_{0.027}^{0.087}$
	Normal	$0.002_{0.001}^{0.011}$	-	$4.715_{4.596}^{4.783}$	$0.837_{0.554}^{1.395}$	$0.033_{0.027}^{0.083}$
	Normal Mixture	$0.002_{0.001}^{0.002}$	-	$4.713_{4.597}^{4.782}$	$0.581_{0.365}^{1.087}$	$0.033_{0.017}^{0.074}$
	2D Normal	$0.004_{0.002}^{0.014}$	-	$4.877_{4.767}^{4.921}$	$250.5_{187.3}^{373.3}$	$0.044_{0.023}^{0.135}$
	2D Normal Mixture	$0.004_{0.004}^{0.016}$	-	$4.875_{4.756}^{4.930}$	$168.4_{122.5}^{238.4}$	$0.042_{0.034}^{0.084}$
	Gamma-Exponential	$0.005_{0.003}^{0.014}$	-	$4.942_{4.907}^{4.985}$	$381.4_{228.4}^{494.7}$	$0.043_{0.023}^{0.094}$
	4D Normal	$0.192_{0.128}^{0.412}$	-	$5.363_{5.304}^{5.409}$	-	$0.093_{0.067}^{0.179}$
	10D Normal	$0.057_{0.044}^{0.132}$	-	$6.960_{6.900}^{6.994}$	-	$1.012_{0.949}^{1.077}$
Mutual Information	2D Normal	$0.011_{0.006}^{0.035}$	-	$10.40_{7.998}^{11.43}$	$224.6_{133.9}^{361.8}$	$0.175_{0.082}^{0.342}$
	2D Normal Mixture	$0.012_{0.008}^{0.044}$	-	$10.27_{9.553}^{11.31}$	$165.2_{94.73}^{362.3}$	$0.170_{0.082}^{0.236}$
	Gamma-Exponential	$0.016_{0.006}^{0.035}$	-	$10.19_{9.919}^{11.59}$	$938.9_{244.7}^{3271.}$	$0.177_{0.141}^{0.256}$
	4D Normal	$0.017_{0.015}^{0.025}$	-	$8.347_{5.171}^{12.02}$	-	$0.601_{0.466}^{1.060}$
	10D Normal	$0.025_{0.024}^{0.040}$	-	$6.232_{6.213}^{6.260}$	-	$3.218_{2.473}^{3.959}$

## Data Availability

The different estimation methods were implemented using Python 3 and are collected in the *UNITE toolbox*, which can be found in the following public repository: https://pypi.org/project/unite-toolbox (accessed on 24 April 2024). The scripts used in the workflow to generate the data for each of the experiments, the evaluation of each estimator on the data for each experiment, and results, as well as the log files can be found in this repository: https://doi.org/10.18419/darus-4087 (accessed on 24 April 2024).

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
