# Peer review of "On the Accurate Estimation of Information-Theoretic Quantities from Multi-Dimensional Sample Data"

_entropy, 2024, doi:10.3390/e26050387_

Round 1

Reviewer 1 Report

Comments and Suggestions for Authors

Please check  the formulas carefully.

Author Response

We sincerely thank the reviewer for their reading of the paper and their comments. We have incorporated their feedback into the revised version of the manuscript.

In the attached PDF we address the reviewer's specific feedback point by point.

Reviewer 2 Report

Comments and Suggestions for Authors

The paper “On the Accurate Estimation of Information-Theoretic Quantities from Multi-Dimensional Sample Data” presents a great review, comparison, and development of the kernel density estimators, binning, and the nearest-neighbor k-NN approaches to evaluate several main measures of information theory on examples of eight different distributions of the dimensionality one, two, four, and ten. The work is theoretically sound, innovative, and practically oriented.

The presentation can be improved in the following aspects.

1.      It is a vast paper of three dozen pages, so, naturally, typos and mistakes can occur, and the whole text should be checked to correct inaccuracies. For instance, the sentence after Table 1 (in p. 10) says “For the simplest 1D cases, there is no dependent variable to calculate mutual information on; so in Cases 1 to 4…”, but the Case 4 in Table 1 is not 1D but already bivariate 2D distribution.

2.      The formulae should be checked as well. For example, the exponent of the normal distribution in the formula (34) (in p. 14) missed the sign “-“.

3.      The precision of estimation for a computational time in Table 2 (in p. 30) is given unnecessary high, up to 7 digits, that should be diminished to increase readability of these results.  

4.      The references are given in amazingly non-careful way, so they should be checked and corrected. In their list, most of them have no year of publication, and books have no place of publishing. The articles have no journal names, sometimes substituted by the link to the net sources, but often without even such a link. For instance, where to find the papers or books given by numbers 9, 24, 34, 46, or 47?  

Resuming, it is a really valuable paper, and subject to the minor revision detailed above the work can be recommended for publication.

Comments on the Quality of English Language

The paper should be checked out for possible typos.

Author Response

We sincerely thank the reviewer for their careful reading of the paper and their positive evaluation. We have incorporated their feedback into the revised version of the manuscript, and provide detailed responses below.

We address their specific comments in the attached PDF.
